# Identifying Partially Observed Causal Models from Heterogeneous/Nonstationary Data

**Xinshuai Dong** [1]  **Haoyue Dai** [1]  **Ignavier Ng** [1]  **Peter Spirtes** [1]  **Kun Zhang** [1,2]

## Abstract

Estimating causal structure in the presence of latent variables is an important yet challenging problem. Recent works have shown that distributional constraints, such as rank deficiency constraints of the covariance matrices, can be exploited to recover the underlying causal structure involving latent variables. However, real-world data often exhibit heterogeneity/nonstationarity, which pose challenges to existing methods. In this work, we develop a principled approach for identifying the structure of partially observed linear causal models from heterogeneous/nonstationary data. We first formulate a class of heterogeneous/nonstationary, partially observed linear causal models and prove that their distributional constraints are equivalent to those in the homogeneous case. Building on this, we propose novel rank tests that can efficiently handle heterogeneous/nonstationary data, and further establish identifiability results for recovering the causal structure involving latent variables. We also provide a method to identify which variables exhibit distribution shifts, i.e., whose causal mechanisms vary across domains. Experiments on simulated and real-world data validate our theoretical findings and the effectiveness of our method (code will be available at https://github.com/dongxinshuai/scm-identify).

## 1. Introduction and Related Work

Discovering causal relations from data is one of the fundamental challenges in many scientific disciplines (Spirtes et al., 2000; Pearl et al., 2000). Traditional causal discovery methods typically assume causal sufficiency, indicating that there are no latent confounders (Spirtes et al., 2000; Spirtes, 2001; Chickering, 2002). However, this assumption may often be violated in practice and ignoring these latent variables can lead to inaccurate causal conclusions. This highlights the importance of causal discovery methods that account for latent confounders.

To address this challenge, early methods such as Fast Causal Inference (FCI) (Spirtes et al., 2000; Zhang, 2008) and its variants (Colombo et al., 2012; Spirtes et al., 2013; Claassen et al., 2013; Akbari et al., 2021) utilize conditional independence tests to identify causal relations among observed variables while accounting for latent confounders. These methods output partial ancestral graphs (PAGs) (Richardson, 1996) over the observed variables, which summarize the equivalence class of causal structures consistent with the data. While FCI does not make any assumption about the latent structure, it often produces less informative outputs, e.g., provides no information about relationships among latent variables. In contrast, recent approaches aim to recover the full causal structure, including latent-to-latent and latent-to-observed relations, by leveraging parametric or graphical assumptions. These approaches are typically based on tetrad or rank constraints (Silva et al., 2003; 2006; Choi et al., 2011; Kummerfeld & Ramsey, 2016; Huang et al., 2022; Dong et al., 2024a), higher-order moments (Shimizu et al., 2009; Cai et al., 2019; Salehkaleybar et al., 2020; Xie et al., 2020; Adams et al., 2021; Dai et al., 2022; Chen et al., 2022; Améndola et al., 2023; Wang & Drton, 2023), matrix decompositions (Anandkumar et al., 2013), and score-based search (Ng et al., 2024).

Apart from latent confounders, another challenge in real-world settings is the presence of heterogeneity in the data. Such variation often arises from different types of interventions, ranging from hard interventions (Eberhardt & Scheines, 2007; Hauser & Bühlmann, 2012) to soft interventions (Eberhardt & Scheines, 2007; Yang et al., 2018). To address this, various constraint-based (Huang et al., 2020), score-based (Hauser & Bühlmann, 2012; Squires et al., 2020; Brouillard et al., 2020), and hybrid (Wang et al., 2017; Yang et al., 2018) methods, as well as other general frameworks (Mooij et al., 2020), have been proposed to infer causal structure from interventional or heterogeneous data.

To handle both latent confounders and heterogeneous/nonstationary data, Magliacane et al. (2016); Kocaoglu et al.

[1]Carnegie Mellon University [2]Mohamed bin Zayed University of Artificial Intelligence.

*Proceedings of the 43rd International Conference on Machine Learning*, Seoul, South Korea. PMLR 306, 2026. Copyright 2026 by the author(s).

(2019) proposed constraint-based methods that rely on conditional independence tests to recover ancestral structures over the observed variables, similar in spirit to FCI. As previously discussed, these outputs may often be uninformative when the goal is to understand the relationships among latent variables. In contrast, a related line of work, causal representation learning (Schölkopf et al., 2021), aims to infer both the latent causal variables and the causal structure among them.

While these methods also leverage interventional or heterogeneous data (Hyvärinen et al., 2023; Ahuja et al., 2023; Squires et al., 2023; von Kügelgen et al., 2023; Zhang et al., 2023; Jin & Syrgkanis, 2023; Zhang et al., 2024; Varıcı et al., 2024a;b; Bing et al., 2024; Ng et al., 2025), they typically make certain assumptions: (1) no causal edges exist among observed variables or from observed to latent variables, and (2) the generative process from latent variables is deterministic (except several works including Khemakhem et al. (2020); Lachapelle et al. (2024)) and invariant across domains. In this work, we consider a more general setting that relaxes these assumptions: we embrace the existence of nonstationarity while allowing all variables, including both observed and latent ones, to be flexibly related. Further discussions of the related works are provided in Section 6. We summarize our contributions as follows:

- We formulate a class of heterogeneous/nonstationary, partially observed linear causal models, i.e., Nonstationary POLCMs (Definition 1), to properly handle nonstationarity; it allows changing model parameters both within and across domains.

- We prove that, despite the existence of nonstationarity, the conditional covariance set generated by Nonstationary POLCMs are equivalent to the covariance set in the homogeneous case (Theorem 1). This implies all the constraints on conditional distribution imposed by structure are equivalent to those in the homogeneous case (Corollary 2), and thus the possibility of those constraint-based homogeneous causal discovery methods to be upgraded to handle the nonstationary scenario.

- To make use of equality constraints for structure identifiability, we establish the relation between rank of conditional covariance and t-separations for Nonstationary POLCMS (Thm. 3); notably, it takes the relation between vanishing partial correlation and d-separations as a special case. To properly control statistical errors with finite data, we further propose two novel statistical tests (Thms. 4 and 5) for the rank of conditional covariance in the nonstationary case, one of which does not require Gaussianity.

- We propose a novel method, LCD-NOD, to identify the structure of Nonstationary POLCMs. The first phase of it serves as a general augmentation of current equality constraint-based methods, e.g., PC (Spirtes et al., 2000),

FOFC (Silva et al., 2003; Kummerfeld & Ramsey, 2016), and RLCD (Dong et al., 2024a), to handle nonstationary data, while the second phase further identifies which variables are directly influenced by the nonstationarity.

## 2. Preliminaries

### 2.1. Problem Setting

To better handle both nonstationarity and latent variables, we assume that data is generated by Nonstationary Partially Observed Linear Causal Models (nonstationary POLCMs), defined as follows.

**Definition 1** (Nonstationary POLCMs). Let $\mathcal{G}$ be a DAG with variable set $\mathbf{V} = \mathbf{X} \cup \mathbf{L} = \{\mathsf{X}_i\}^n \cup \{\mathsf{L}_i\}^m$ that contains $n$ observed and $m$ latent variables. Each variable $\mathsf{V}_i \in \mathbf{V}$ is generated following

$$\mathsf{V}_i = \sum\nolimits_{\mathsf{V}_j \in Pa(\mathsf{V}_i)} h_{j,i}(\mathsf{T}, \delta_{j,i})\mathsf{V}_j + g_i(\mathsf{T}, \epsilon_i), \quad (1)$$

where $Pa(\mathsf{V}_i)$ denotes the parent set of $\mathsf{V}_i$, $h_{j,i}(\mathsf{T}, \delta_{j,i})$ denotes the edge coefficient from $V_j$ to $V_i$, and $\delta_{j,i}$ and $\epsilon_i$ are independent noise terms.

In Definition 1, $\mathsf{T}$ can be understood as the domain index. The coefficient for edge $\mathsf{V}_j \to \mathsf{V}_i$ is $h_{j,i}$, which is a deterministic function of $\mathsf{T}$ and $\delta_{j,i}$. The additive noise term $g_i$ is also a deterministic function of $\mathsf{T}$ and $\epsilon_i$. Therefore, in Equation (1), two kinds of nonstationarity can be modeled. (i) Nonstationarity across domains, as both $h_{j,i}$ and $g_i$ are functions of domain index $\mathsf{T}$. (ii) Nonstationarity within domain, as edge coefficients $h_{j,i}$ is also a function of independent noise term $\delta_{j,i}$. It is possible to model these two kinds of nonstationarity in a more complex functional fashion, and yet one major goal of this work is to show that the constraints are equivalent to those in the homogeneous case; we conjecture that any further relaxation of functional forms would induce failure of Theorem 1 (detailed in Section 3.2). A more detiled discussion on the rationale behind can be found in Section C.1.

Given i.i.d. samples of observed variables $\mathbf{X}$ generated by Def. 1, our objective is to identify the causal structure of the underlying nonstationary causal model. More specifically, we aim to identify the causal structure among all the variables, including both observed and latent ones. Further, the generating process for some variables might be stationary. Thus we are also interested in identifying the stationary and nonstationary variable set, or equivalently, identifying $\mathcal{G}^{\text{aug}}$, where $\mathcal{G}^{\text{aug}}$ is the augmented graph of $\mathcal{G}$ such that $\mathcal{G}^{\text{aug}}$ contains one additional node $\mathsf{T}$ and $\mathsf{T} \to \mathsf{V}_i$ if and only if $h_{:,i}$ and $g_i$ can change with different values of $\mathsf{T}$.

### 2.2. Notations and Paper Organization

In this paper, we use $\mathsf{V}$ to denote random variable and $\mathbf{V}$ a set of random variables. We use $\mathsf{T}$ to refer to the ran-

*Table 1.* Graphical notations used throughout this paper.

| Pa: Parents | $\mathbf{V}$: Variables | V: Variable | T: Domain / Time index |
|---|---|---|---|
| Ch: Children | $\mathbf{L}$: Latent variables | L: Latent variable | $\top$: Matrix transpose |
| PCh: Pure children | $\mathbf{X}$: Observed variables | X: Observed variable | $t$: value of variable T |
| $\mathcal{G}$: Ground truth structure | $\mathcal{G}^{\mathrm{aug}}$: Structure including T | $\boldsymbol{D}^{\mathbf{V}}$: Data matrix of $\mathbf{V}$ | $\boldsymbol{D}^{\mathbf{V}|t}$: Data of $\mathbf{V}|\mathsf{T} = t$ |

dom variable that represents domain index, $\top$ the matrix transpose, and $t$ the observed value of T. A summary of commonly used notations can be found in Table 1.

The rest of the paper is organized as follows. In Section 3.1, we briefly introduce the constraints in the homogeneous setting for structure identification and show that they normally fail to hold in the nonstationary scenario. In Section 3.2, we formally show the equivalence relation between constraints on the covariance in the homogeneous case and the constraints on the conditional covariance in the nonstationary case. In Section 3.3, we focus on rank constraints and characterize their graphical implications for structure identification, with two novel rank tests of conditional covariance proposed in Section 3.4. In Section 4, we propose the Latent variable Causal Discovery from heterogeneous/NOnstationary Data (LCD-NOD) algorithm, and empirically validate LCD-NOD in Section 5.

## 3. Distributional Information for Structure Identification

### 3.1. Constraints in the Homogeneous Case

In this section, we first revisit the constraints in homogeneous Partially Observed Linear Causal Models (POLCMs), defined as follows,

**Definition 2** (Homogeneous POLCMs). Let $\mathcal{G}$ be a DAG with variable set $\mathbf{V} = \mathbf{X} \cup \mathbf{L} = \{\mathsf{X}_i\}^n \cup \{\mathsf{L}_i\}^m$ that contains $n$ observed and $m$ latent variables. Each variable $\mathsf{V}_i \in \mathbf{V}$ is generated following

$$\mathsf{V}_i = \sum\nolimits_{\mathsf{V}_j \in Pa(\mathsf{V}_i)} f_{j,i} \mathsf{V}_j + \epsilon_i, \qquad (2)$$

where $Pa(\mathsf{V}_i)$ denotes the parents of $\mathsf{V}_i$, $f_{j,i}$ the edge coefficient from $\mathsf{V}_j$ to $\mathsf{V}_i$, and $\epsilon_i$ the independent noise terms.

For a model in Definition 2, its structure $\mathcal{G}$ imposes various constraints on the generated population covariance matrices, regardless of the parameter values (i.e., $(f_{j,i})$ and variance of $\epsilon_i$). These constraints have two categories: equality and inequality constraints. For structure identification, equality constraints are more useful (Dong et al., 2026) and they includes conditional independence (or vanishing partial correlation) constraints (Spirtes et al., 2000), which suffice to identify the whole structure up to the Markov Equivalence Class (MEC) in the absence of latent variables, as special cases. To handle latent variables, more equality constraints have been discovered and exploited, such as rank constraints

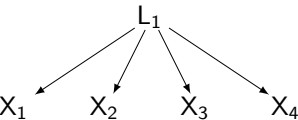

*(a)* $\mathcal{G}$ under the homogeneous POLCMs setting, where the edge coefficients and the variance of the noise terms are fixed.

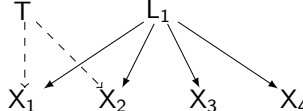

*(b)* $\mathcal{G}^{\mathrm{aug}}$ under nonstationary setting where the dashed arrows from T to $\mathsf{X}_1$ and $\mathsf{X}_2$ means the edge coefficients from $\mathsf{L}_1$ to $\mathsf{X}_1$ and $\mathsf{X}_2$ change across T.

*Figure 1.* With the same structure but in the presence of nonstationarity, the same equality constraint does not hold anymore. Specifically, in (a) $\sigma_{\mathsf{X}_1,\mathsf{X}_2}\sigma_{\mathsf{X}_3,\mathsf{X}_4} = \sigma_{\mathsf{X}_1,\mathsf{X}_3}\sigma_{\mathsf{X}_2,\mathsf{X}_4}$ holds regardless of the choice of parameters, while in (b) it does not.

/ vanishing determinant (Sullivant et al., 2010) and Verma constraints (Verma & Pearl, 1991). An overview can be found in (Drton, 2018).

Given that these equality constraints contain crucial information for structure identifiability, a question naturally arises: Can these constraints in the homogeneous case be directly extended to the nonstationary setting? Unfortunately, they do not generally carry over, as illustrated as follows.

**Example 1.** *In Figure 1, the model in (a) follows Definition 2 and the model in (b) follows Definition 1, but they share the same DAG structure among $\mathbf{V}$. However, the equality constraints in (a) may not hold any more in (b). For example, the classical tetrad constraint (a kind of equality constraint) implies $\sigma_{\mathsf{X}_1,\mathsf{X}_2}\sigma_{\mathsf{X}_3,\mathsf{X}_4} = \sigma_{\mathsf{X}_1,\mathsf{X}_3}\sigma_{\mathsf{X}_2,\mathsf{X}_4}$ in (a) regardless of the choice of parameter in (a), as $\frac{\sigma_{\mathsf{X}_1,\mathsf{X}_2}\sigma_{\mathsf{X}_3,\mathsf{X}_4}}{\sigma_{\mathsf{X}_1,\mathsf{X}_3}\sigma_{\mathsf{X}_2,\mathsf{X}_4}} = \frac{\mathbb{E}[f_{\mathsf{L}_1,\mathsf{X}_1}f_{\mathsf{L}_1,\mathsf{X}_2}]\mathbb{E}[f_{\mathsf{L}_1,\mathsf{X}_3}f_{\mathsf{L}_1,\mathsf{X}_4}]}{\mathbb{E}[f_{\mathsf{L}_1,\mathsf{X}_1}f_{\mathsf{L}_1,\mathsf{X}_3}]\mathbb{E}[f_{\mathsf{L}_1,\mathsf{X}_2}f_{\mathsf{L}_1,\mathsf{X}_4}]} = 1$ always holds due to that $(f_{j,i})$ are constants. However, in (b) $\frac{\sigma_{\mathsf{X}_1,\mathsf{X}_2}\sigma_{\mathsf{X}_3,\mathsf{X}_4}}{\sigma_{\mathsf{X}_1,\mathsf{X}_3}\sigma_{\mathsf{X}_2,\mathsf{X}_4}} = \frac{\mathbb{E}_{\mathsf{T},\delta,\epsilon}[h_{\mathsf{L}_1,\mathsf{X}_1}h_{\mathsf{L}_1,\mathsf{X}_2}]\mathbb{E}_{\mathsf{T},\delta,\epsilon}[h_{\mathsf{L}_1,\mathsf{X}_3}h_{\mathsf{L}_1,\mathsf{X}_4}]}{\mathbb{E}_{\mathsf{T},\delta,\epsilon}[h_{\mathsf{L}_1,\mathsf{X}_1}h_{\mathsf{L}_1,\mathsf{X}_3}]\mathbb{E}_{\mathsf{T},\delta,\epsilon}[h_{\mathsf{L}_1,\mathsf{X}_2}h_{\mathsf{L}_1,\mathsf{X}_4}]}$ where the expectation is taken over $\mathsf{T}, \delta, \epsilon$ and both $h_{\mathsf{L}_1,\mathsf{X}_1}$ and $h_{\mathsf{L}_1,\mathsf{X}_2}$ are functions of $\mathsf{T}, \delta$, and thus the equality constraint does not generally hold.*

By the above example, we see that the equality constraints does not readily transfer to the nonstationary case. Thus we need to characterize the equality constraints in the nonstationary case for structure identification, detailed as follows.

## 3.2. Constraints Implied by Structure Under Nonstationarity

In this section, we aim to characterize useful constraints in the nonstationary scenario for structure identification. To this end, we need to first define the observational covariance set under POLCMs, as $\Theta(\mathcal{G})$, and the observational conditional covariance set under Nonstationary POLCMs, as $\Phi(\mathcal{G})$, in Definition 3 and Definition 4, respectively. The reason why we care about $\Theta(\mathcal{G})$ and $\Phi(\mathcal{G})$ is as follows. A constraint imposed by $\mathcal{G}$ on the covariance matrix is nothing but some relations among the entries of the covariance matrix that always holds regardless of the choice of the parameters; Therefore, constraints imposed by $\mathcal{G}$ under POLCMs and Nonstationary POLCMs are just properties of $\Theta(\mathcal{G})$ and $\Phi(\mathcal{G})$ respectively.

**Definition 3** (Observational covariance set under POLCMs). Let $F = (f_{j,i})$ and $\Omega$ be the covariance matrix for $\{\epsilon_i\}^{n+m}$. We define the observational covariance set of $\mathcal{G}$ under POLCMs (Definition 2) as:

$$\Theta(\mathcal{G}) := \{\Theta : \Theta = ((I - F^\top)^{-1}\Omega(I - F^\top)^{-\top})_{[:n,:n]},$$

$$\text{for any } (F, \Omega) \text{ s.t. } \Omega \in \text{diag}^+ \text{ and } \text{supp}(F) \subseteq \text{supp}(F_\mathcal{G})\}. \quad (3)$$

$\Theta(\mathcal{G})$ simply refers to the set of all possible covariance matrices over $\{X_i\}^n$, by taking the same causal structure $\mathcal{G}$ and different model parameters $(F, \Omega)$. Note that if we do not further assume any non-Gaussianity for $\epsilon_i$ in Definition 2 (which is often beneficial for structure identifiability), then only the second-order information matters, and thus all the distributional information that are useful for structure identification are just all the properties of $\Theta(\mathcal{G})$.

**Definition 4** (Observational conditional covariance set under Nonstationary POLCMs). Let $H(\mathsf{T}) = (h_{j,i}(\mathsf{T}, \delta_{j,i}))$ and $g(\mathsf{T}) = (g_i(\mathsf{T}, \epsilon_i))$. The observational conditional covariance set of $\mathcal{G}$ under Nonstationary POLCMs (Definition 1) is defined as:

$$\Phi(\mathcal{G}) := \{\Phi : \Phi = \mathbb{E}_{p(\mathbf{V}|\mathsf{T})}[(I - H(\mathsf{T})^\top)^{-1}g(\mathsf{T})g(\mathsf{T})^\top$$

$$(I - H(\mathsf{T})^\top)^{-\top}]_{[:n,:n]}, \text{for any } (H(\mathsf{T}), g(\mathsf{T}))$$

$$\text{s.t. } \text{supp}(H(\mathsf{T})) \subseteq \text{supp}(H_\mathcal{G})\}. \quad (4)$$

Similar to $\Theta(\mathcal{G})$, $\Phi(\mathcal{G})$ is the set of all possible *conditional* covariance matrices over $\{X_i\}^n$, by taking the same $\mathcal{G}$ and different model parameters $(H, g)$. We note that the definition of $\Phi(\mathcal{G})$ involve conditioning on $\mathsf{T}$, but for any specific value of $\mathsf{T}$, $\Phi(\mathcal{G})$ keeps the same, and thus we omit the notion $\mathsf{T}$ in $\Phi(\mathcal{G})$.

Given the definitions of observational covariance set $\Theta(\mathcal{G})$ and observational conditional covariance set under nonstationarity $\Phi(\mathcal{G})$, we now introduce the main theoretical result of this section in what follows.

**Theorem 1** (Equivalence Relation between $\Theta(\mathcal{G})$ under POLCMs and $\Phi(\mathcal{G})$ under Nonstationary POLCMs). *Given a DAG $\mathcal{G}$, let $\Theta(\mathcal{G})$ be the observational covariance set of $\mathcal{G}$ under POLCMS, as in Definition 3, and let $\Phi(\mathcal{G})$ be the observational conditional covariance set of $\mathcal{G}$ under Nonstationary POLCMs, as in Definition 4. We have that* $\Theta(\mathcal{G}) = \Phi(\mathcal{G})$.

Theorem 1 says that, given the same structure $\mathcal{G}$, despite the presence of both inter-domain and intra-domain nonstationarity in Nonstationary POLCMs, the covariance set in the homogeneous case is exactly the same as the conditional covariance set in the nonstationary case. As constraints are just properties of the covariance / conditional covariance set, it can be inferred from Theorem 1 that all the constraints on $\Theta(\mathcal{G})$ are exactly the same as those on $\Phi(\mathcal{G})$, formalized in Corollary 2.

**Corollary 2** (Equality and Inequality Constraints Under Nonstationary). *If $\mathcal{G}$ implies an equality or inequality constraint on $\Theta(\mathcal{G})$, then $\mathcal{G}$ also implies the same constraint on $\Phi(\mathcal{G})$, and vice versa.*

Theorem 1 and Corollary 2 are significant in the sense that they show the possibility of structure identifiability of Nonstationary POLCMs. The key takeaways are two folds.

- All the constraints implied by $\mathcal{G}$ on $\Theta(\mathcal{G})$ remains on $\Phi(\theta)$; Thus all the equality-constraint-based homogeneous causal discovery methods, e.g., PC (partial correlation) (Spirtes et al., 2000), FOFC (Tetrad) (Silva et al., 2003; Kummerfeld & Ramsey, 2016), and RLCD (rank constraints) (Dong et al., 2024a), can be upgraded to handle the nonstationary scenario (detailed in Section 4).

- In Definition 4, we do not restrict the function forms of $h, g$ or the distributions of $\delta, \epsilon$, and thus we cannot assume that the joint distribution over $\mathbf{X}|\mathsf{T}$ to be non-Gaussian (which is however beneficial to structure identifiability as in Shimizu et al. (2006)). As a consequence, only the second-order information can be reliably used and thus those constraints on $\Phi(\mathcal{G})$ are all the graphical information that we can reliably use from the observed conditional distribution $p(\mathbf{X}|\mathsf{T})$. In other words, we focus on the constraints on $\Phi(\mathcal{G})$ for building asymptotic structure identifiablity, and the identifiability theory requires neither Gaussianity nor non-Gaussianity assumption.

### 3.3. Rank Constraint under Nonstationarity and its Graphical Implication

In Section 3.2, we have shown the equality constraints in the homogeneous case can be transferred to the constraints in the nonstationary case. In this section, we focus on a specific class of equality constraints, rank deficiency constraints (Sullivant et al., 2010), to establish its relation with

trek-separations (definition of trek in Definition 7 while definition of trek-separation in Definition 5) and thus the structure identifiability for Nonstationary POLCMs.

The reason why we focus on rank constraints are as follows. (i) Rank constraints imply t-separations (Sullivant et al., 2010), and thus contain useful graphical information about latent variables. By making use of rank constraints, we can employ RLCD algorithm (Dong et al., 2024a) to identify latent variable structures where all variables, including both observed and latent ones, can be very flexibly related. Thus, when rank constraints are properly characterized in the nonstationary setting, we can elegantly upgrade RLCD to identify the structure of Nonstationary POLCMs. (ii) The set of all rank constraints is itself a very large subset of the set of all equality constraints. In fact, rank constraints take both vanishing partial correlation constraints and Tetrad constraints as special cases (Sullivant et al., 2010; Dong et al., 2024a). That is to say, when rank constraints are properly characterized, we can translate the vanishing partial correlation constraints and Tetrad constraints into rank constraints, and thus readily upgrade the PC algorithm (Spirtes et al., 2000) (based on vanishing partial correlation) and the FOFC for One Factor Models (Silva et al., 2003) (based on Tetrad).

**Definition 5** (T-separation (Sullivant et al., 2010)). Let $\mathbf{A}$, $\mathbf{B}$, $\mathbf{C_A}$, and $\mathbf{C_B}$ be four set of variables in $\mathcal{G}$ (not necessarily disjoint). $(\mathbf{C_A}, \mathbf{C_B})$ t-separates $\mathbf{A}$ from $\mathbf{B}$ if for every trek $(P_1, P_2)$ from a vertex in $\mathbf{A}$ to a vertex in $\mathbf{B}$, either $P_1$ contains a vertex in $\mathbf{C_A}$ or $P_2$ contains a vertex in $\mathbf{C_B}$.

Below we introduce the main theoretical result in this section: rank constraints for Nonstationary POLCMs and its graphical implication of t-separations. This result, to our best knowledge, is the first extension of the characterization of rank and t-separations to the nonstationary setting.

**Corollary 3** (Rank and T-separation for Nonstationary POLCMs). *Given two sets of variables $\mathbf{A}$ and $\mathbf{B}$ generated by Nonstationary POLCMs with DAG $\mathcal{G}$ and assume rank faithfulness (Spirtes, 2013; Dong et al., 2024a). We have:*

$$rank(\Sigma_{\mathbf{A},\mathbf{B}|\mathsf{T}}) = \min\{|\mathbf{C_A}| + |\mathbf{C_B}| : (\mathbf{C_A}, \mathbf{C_B})$$
$$t\text{-}separates\,\mathbf{A}\,from\,\mathbf{B}\,in\,\mathcal{G}\}, \quad (5)$$

*where $\Sigma_{\mathbf{A},\mathbf{B}|\mathsf{T}}$ is the cross-covariance over $\mathbf{A}$ and $\mathbf{B}$ conditional on $\mathsf{T}$.*

### 3.4. Statistical Rank Tests for Conditional Covariance

In Sec. 3.3, we formally characterize the relation between rank and t-separations for Nonstationary POLCMs. Note that Corollary 3 requires population conditional covariance, and yet in real-life applications we only have access to finite samples and thus the empirical counterpart. To properly control statistical errors, we need a valid statistical test to test the rank of the underlying population conditional covariance, by making use of finite i.i.d. observations.

A straight forward way is to just use the classical rank test (Bartlett, 1947; Camba-Méndez & Kapetanios, 2009) (which assumes joint Gaussianity), by using data conditioned on a value of $\mathsf{T}$. E.g., assume we are given i.i.d. observations of $\mathbf{X}$ and $\mathsf{T}$, where $\mathsf{T} \in \{1, 2\}$. We can just pick the data points such that $\mathsf{T} = 1$, and use these samples to do classical rank tests. The problem of this straight forward method is that, when the number of domains is large, we only make use of a very small portion of data. E.g., assume that $\mathsf{T} \in \{1, ..., 100\}$ with uniform distribution and we are given 1000 data points. This straight forward method has to condition on a value of $\mathsf{T}$ and thus only makes use of 10 data points for the test. Thus, a valid test that can utilize all the data points simultaneously would be essential.

To this end, we propose two novel statistical tests for the rank of conditional covariance matrix, and both of them can make use of data from all the domains. The first one, formalized in Theorem 4, is based on likelihood ratio statistics by assuming Gaussianity for the parametric form of likelihood, as follows.

**Theorem 4** (Likelihood Ratio Statistics for Rank of Conditional Covariance). *Given two sets of random variables $\mathbf{A}$ and $\mathbf{B}$ with $|\mathbf{A}| = P$, $|\mathbf{B}| = Q$, $K = \min(P, Q)$ and $\mathbf{A} \cup \mathbf{B}$ are jointly gaussian given $\mathsf{T}$. Assume the null hypothesis $\mathcal{H}_0$ is $rank(\Sigma_{\mathbf{A},\mathbf{B}|\mathsf{T}}) \leq k$, the likelihood ratio statistics $\Lambda(k)$ is*

$$\sum_{t \in supp(\mathsf{T})} \frac{P + Q + 1 - 2N_t}{2} \ln(\prod_{i=k+1}^{K} (1 - r_{t,i}^2)), \quad (6)$$

*where $N_t$ is the number of data points such that $\mathsf{T} = t$, and $r_{t,i}$ is the i-th canonical correlation between $\mathbf{A}$ and $\mathbf{B}$ conditioned on $\mathsf{T} = t$. Further, $\Lambda(k)$ converges in distribution to $\chi_{df}^2$, with degree of freedom $\sum_{t \in supp(\mathsf{T})}(P - k)(Q - k)$.*

To perform the test, we just calculate the test statistics $\Lambda(k)$ and plug it in to the corresponding chi-square distribution to get the p-value. As in Thm. 4 and its proof, $\Lambda(k)$ is a likelihood ratio statistics based on $p(\mathbf{A} \cup \mathbf{B}, \mathsf{T})$ instead of $p(\mathbf{A} \cup \mathbf{B}|\mathsf{T})$, and thus it makes use of all the data points instead of conditioning on a specific value of $\mathsf{T}$. Further, as a likelihood ratio test, its asymptotic optimality in terms of power can normally be guaranteed under regularity conditions (Van der Vaart, 2000; Lehmann et al., 1986).

For the test, if we do not assume the parametric form of the likelihood as Gaussian, the same statistics can still be employed, but the asymptotic null distribution is not necessarily chi-square anymore. In this case, we can rely on data permutation to empirically estimate the null distribution and thus correctly calculate the p-value. Specifically, the second test that we propose for conditional covariance rank does not require jointly Gaussianity of $\mathbf{A} \cup \mathbf{B}$ given $\mathsf{T}$ at all, detailed in Theorem 5 as follows.

**Theorem 5** (Permutation-based Rank Test of Conditional Covariance). *Given random vectors $\mathbf{A}$ and $\mathbf{B}$ with $|\mathbf{A}| = P, |\mathbf{B}| = Q$ and their corresponding data matrices $\boldsymbol{D}^{\mathbf{A}|t} \in \mathbb{R}^{N_t \times P}$ and $\boldsymbol{D}^{\mathbf{B}|t} \in \mathbb{R}^{N_t \times Q}$ for all $t \in supp(\mathsf{T})$ generated following Definition 1, and the null hypothesis $\mathcal{H}_0$ as $rank(\Sigma_{\mathbf{A},\mathbf{B}|\mathsf{T}}) \leq k$, the following test is valid.*

*(i) For each $t \in supp(\mathsf{T})$, solve the following CCA problem by SVD: $\boldsymbol{USV} = \hat{\Sigma}_{\mathbf{A}|t}^{-\frac{1}{2}} \hat{\Sigma}_{\mathbf{A},\mathbf{B}|t} \hat{\Sigma}_{\mathbf{B}|t}^{-\frac{1}{2}}$, $\boldsymbol{R}^{\mathbf{A}} = \hat{\Sigma}_{\mathbf{A}|t}^{-\frac{1}{2}\top} \boldsymbol{U}$, and $\boldsymbol{R}^{\mathbf{B}} = \hat{\Sigma}_{\mathbf{B}|t}^{-\frac{1}{2}\top} \boldsymbol{V}^{\top}$, and define canonical vectors $\mathbf{C}_{\mathbf{A}|t} = \boldsymbol{R}^{\mathbf{A}\top} \mathbf{A}|t$ and $\mathbf{C}_{\mathbf{B}|t} = \boldsymbol{R}^{\mathbf{B}\top} \mathbf{B}|t$, with data matrices as $\boldsymbol{D}^{\mathbf{C}_{\mathbf{A}}|t} = \boldsymbol{D}^{\mathbf{A}|t} \boldsymbol{R}^{\mathbf{A}}$ and $\boldsymbol{D}^{\mathbf{C}_{\mathbf{B}}|t} = \boldsymbol{D}^{\mathbf{B}|t} \boldsymbol{R}^{\mathbf{B}}$, respectively.*

*(ii) Define a set of random permutation matrices $\mathcal{P} = \{\boldsymbol{P}_t : t \in supp(\mathsf{T}), \boldsymbol{P}_t \text{ is a } N_t \times N_t \text{ permutation matrix}\}$.*

*(iii) Define the statistics under each $\mathcal{P}$, $\Lambda(k, \mathcal{P})$, as*

$$\sum_{t \in supp(\mathsf{T})} \frac{P + Q + 1 - 2N_t}{2} \ln\left(\prod_{i=1}^{K-k} (1 - \hat{r}_{t,\boldsymbol{P}_t,i}^2)\right) \quad (7)$$

*where $\hat{r}_{t,\boldsymbol{P}_t,i}^2$ refers to the $i$-th CCA score between data matrices $\boldsymbol{P}_t \boldsymbol{D}_{:,k:}^{\mathbf{C}_{\mathbf{A}}|t}$ and $\boldsymbol{D}_{:,k:}^{\mathbf{C}_{\mathbf{B}}|t}$.*

*(iv) Calculate the p-value as $p_k = \mathbb{E} \, \mathbf{1}_{[\Lambda(k,\mathcal{P}) \geq \Lambda(k,\mathcal{I})]}$, where the expectation is taken over random $\mathcal{P}$ and $\mathcal{I} = \{\boldsymbol{I}_t : t \in supp(\mathsf{T}), \boldsymbol{I}_t \text{ is the } N_t \times N_t \text{ identity matrix}\}$.*

Roughly, the permutation test described in Theorem 5 makes use of zero correlation between canonical variables given $t$ under the null hypothesis (which is partly inspired by Dong et al. (2025)) to establish the exchangeability, and then resample each term related to $t$ by permutation in order to resample from the global statistics $\Lambda(k, \mathcal{I})$ (detailed in the proof) to correctly calculate the p-value. Thus it does not require Gaussianity at all.

We empirically validate the proposed two tests by our experiments in Section B.1. It is worth noting that both tests can control the Type-I properly and control the Type-II better than baselines, under both Gaussian and non-Gaussian settings, even though the first test theoretically requires Gaussianity (Figures 8 to 11).

# 4. Latent Variable Causal Discovery from NOnstationary Data

In this section, we present LCD-NOD (Latent variable Causal Discovery from heterogeneous/NOnstationary Data), a two-phase algorithm that first recovers the graph $\mathcal{G}$ from non-stationary data, and then identifies changes across domains, represented by the augmented graph $\mathcal{G}^{\text{aug}}$.

## 4.1. Phase 1: Identification of Structure $\mathcal{G}$

The first phase of LCD-NOD aims to identify the causal structure $\mathcal{G}$ among all the variables $\mathbf{V} = \mathbf{X} \cup \mathbf{L}$. It is

designed as a general augmentation of existing equality-constraint-based methods to work in the nonstationary scenario. The key idea is simple and effective: for a given causal discovery algorithm, replace the test of equality constraint on the covariance matrices by the proposed conditional rank tests. Next, we take PC and RLCD as two examples to show how Phase 1 works.

**(i)** For PC algorithm (Spirtes et al., 2000) (and those based on vanishing partial correlations such as FCI (Spirtes et al., 2000; Zhang, 2008)), replace the test of $\mathsf{V}_1 \perp\!\!\!\perp \mathsf{V}_2 | \mathbf{V}_3$ by the rank test using Theorem 4. Specifically, if we fail to reject $rank(\Sigma_{\{\mathsf{V}_1\} \cup \mathbf{V}_3, \{\mathsf{V}_2\} \cup \mathbf{V}_3 | \mathsf{T}}) \leq |\mathbf{V}_3|$, then we fail to reject $\mathsf{V}_1 \perp\!\!\!\perp \mathsf{V}_2 | \mathbf{V}_3$. **(ii)** For rank based methods, e.g., Hier-Rank (Huang et al., 2022) and RLCD (Dong et al., 2024a), replace the test of $rank(\Sigma_{\mathbf{V}_1, \mathbf{V}_2}) \leq k$ by the rank test using Theorem 4. Specifically, if we fail to reject $rank(\Sigma_{\mathbf{V}_1, \mathbf{V}_2 | \mathsf{T}}) \leq k$, then we fail to reject $rank(\Sigma_{\mathbf{V}_1, \mathbf{V}_2}) \leq k$. Further, for those methods that are based on Tetrad constraints, we can just reformulate the Tetrad constraint into a rank constraint and test it using Theorem 4 or Theorem 5.

LCD-NOD Phase 1 serves as a general framework in which any existing latent variable causal discovery method based on covariance information (e.g., for linear Gaussian models) can be directly generalized to handle nonstationary data. This is justified by Theorem 1 which shows that in each domain, the induced covariance set remains identical to that of a static model despite nonstationarity.

We conclude introducing LCD-NOD Phase 1 by formally stating its structure identifiability:

**Theorem 6** (Structure Identifiability of LCD-NOD Phase 1). *Given an equality-constraint-based causal discovery method $\mathcal{M}$, if $\mathcal{M}$ asymptotically identifies $\mathcal{G}$ up to equivalence class $\mathcal{C}$ under POLCMs and graphical assumption $\mathcal{A}$, then the Phase 1 of LCD-NOD, i.e., the augmented $\mathcal{M}$, asymptotically identifies $\mathcal{G}$ up to $\mathcal{C}$, under Nonstationary POLCMs and $\mathcal{A}$.*

## 4.2. Phase 2: Identification of Augmented Structure $\mathcal{G}^{\text{aug}}$

In previous sections, we show that any existing latent variable causal discovery method based on covariance information can be directly applied using each single domain's information. In this section, we go further by leveraging data across different domains to identify where changes happen. For a case study, we consider a specific model, the one-factor model (Silva et al., 2003).

One-factor model captures cases where latent variables are indirectly measured. By introducing nonstationarity, it becomes a special case of the Nonstationary POLCM model (Definition 1) as follows:

**Definition 6** (Nonstationary one-factor model). Let $\mathcal{G}$ be a DAG over latent variables $\mathbf{L} = \{\mathsf{L}_i\}^m$ where each $\mathsf{L}_i$ is generated following $\mathsf{L}_i = \sum_{\mathsf{L}_j \in Pa(\mathsf{L}_i)} h_{j,i}(\mathsf{T}, \delta_{j,i})\mathsf{L}_j +$

$g_i(\mathsf{T}, \epsilon_i)$. Each latent variable $\mathsf{L}_i$ is then associated with a set $\mathbf{X}_i$ of at least two observed variables (i.e., $|\mathbf{X}_i| \geq 2$) as its pure measurements. For each pure measurement $\mathsf{X}_i^{(k)} \in \mathbf{X}_i$, it is generated by $\mathsf{X}_i^{(k)} = h'_{i,k}(\mathsf{T}, \delta'_{i,k})\mathsf{L}_i + g'_{i,k}(\mathsf{T}, \epsilon'_{i,k})$, where we use primed notation (e.g., $h'$) to distinguish parameters in the measurement process from those governing the latent variable dynamics.

Note that in Phase 1, although the latent variables themselves are unobserved, the CI relations among them manifest as testable low-rank in observed measurements, due to the equivalent linearity in each domain (Theorem 6). In Phase 2, however, this rank-based correspondence breaks down, as changes to latent variables–represented by edges $\mathsf{T} \rightarrow \mathsf{L}_i$–can induce complex, nonlinear dependencies. Hence, instead of testing for CI relations between $\mathsf{T}$ and $\mathbf{L}$, we take a different route: parameter identification. We show that model parameters can be identified up to trivial indeterminacies in each domain, allowing us to detect changes by comparing identified parameters. We formalize this below.

We first introduce the model identification result. In each domain, there are following unknown model parameters: $\Sigma_{\mathbf{L},\mathbf{L}} \in \mathbb{R}^{m \times m}$, the variance-covariance matrix among $\mathbf{L}$; $\omega^{(1)}, \omega^{(2)}, \phi^{(1)}, \phi^{(2)} \in \mathbb{R}^m$, the equivalent linear coefficients and exogenous noise variances of each latent variable's two pure measurements, where for $k = 1, 2$, $\omega_i^{(k)} = \mathbb{E}[h'_{i,k}(t, \delta'_{i,k})]$, and $\phi_i^{(k)} = \mathrm{Var}[g'_{i,k}(t, \epsilon'_{i,k})]$. Let $\hat{\Sigma}_{\mathbf{L},\mathbf{L}}, \hat{\omega}^{(1)}, \hat{\omega}^{(2)}, \hat{\phi}^{(1)}, \hat{\phi}^{(2)}$ be another set of (estimated) parameters that yield the same covariance for $\mathbf{X}$. Then, the true parameters are identified up to scaling, i.e., $\frac{\hat{\omega}^{(1)}}{\omega^{(1)}} = \frac{\hat{\omega}^{(2)}}{\omega^{(2)}} =: c \in \mathbb{R}^m$, where the division is element-wise, and $\Sigma_{\mathbf{L},\mathbf{L}} = \mathrm{diag}(c) \hat{\Sigma}_{\mathbf{L},\mathbf{L}} \mathrm{diag}(c)$ holds. Moreover, measuring noise variances are identified, i.e., $\phi^{(k)} = \hat{\phi}^{(k)}$, $k = 1, 2$. Proofs and solution details are given in Section A.

With the parameter identification results in each domain, we proceed to identify changes by comparing the recovered parameters across domains. Since latent variables are not accessible, this comparison must come with a trade-off. Two levels of assumptions are needed–one to address scaling indeterminacies, and one to ensure faithfulness so that changes leave trace in the second-order information:

**Assumption 7.** *Two assumptions are needed to identify changes from recovered model parameters:*

*(A1.)* *To address indeterminacies, for each latent variable $\mathsf{L}_i$, at least one of its measurement's equivalent linear coefficient, w.l.o.g. assumed to be $\omega_i^{(1)}$, remains invariant across domains.*

*(A2.)* *To ensure faithfulness, if the generating process of a latent variable $\mathsf{L}_i$ changes, then there exist at least two domains in which, for any subset $\mathbf{L}_C \subseteq \mathbf{L} \setminus \{\mathsf{L}_i\}$, the conditional variance of $\mathsf{L}_i$ given $\mathbf{L}_C$ (calculated*

*from $\Sigma_{\mathbf{L},\mathbf{L}}$) differs. Similarly, if a measurement variable $\mathsf{X}_i^{(k)}$ is changed, its corresponding exogenous noise variance $\phi_i^{(k)}$ must change as well.*

Due to space limit, we leave the explanation and justification of assumptions to Theorem 8. Under the assumptions, we can now formally state the result for identifying changes, as in Thm. 8, to get the links from $\mathsf{T}$ to $\mathbf{L}$ and $\mathbf{X}$. After that, further orientation can be done by using e.g., Meek rules.

**Theorem 8** (Identification of changing variables). *Suppose measurements $\mathbf{X}$ are generated following Definition 6 and let $\{\hat{\Sigma}_{\mathbf{L},\mathbf{L}|t}, \hat{\omega}_t^{(1)}, \hat{\omega}_t^{(2)}, \hat{\phi}_t^{(1)}, \hat{\phi}_t^{(2)}\}$ be the model parameters estimated in each domain $\mathsf{T} = t$. Denote the normalized latent covariance matrices by $\{\hat{\Sigma}'_{\mathbf{L},\mathbf{L}|t} := \mathrm{diag}(\hat{\omega}_t^{(1)}) \hat{\Sigma}_{\mathbf{L},\mathbf{L}|t} \mathrm{diag}(\hat{\omega}_t^{(1)})\}$. Then, under (A1) and (A2), $\mathsf{T} \rightarrow \mathsf{L}_i \in \mathcal{G}^{aug}$ if and only if for all subsets $\mathbf{L}_C \subseteq \mathbf{L} \setminus \{\mathsf{L}_i\}$, the conditional variances $\{\hat{\mathrm{Var}}'_t(\mathsf{L}_i|\mathbf{L}_C)\}$ calculated from $\{\hat{\Sigma}'_{\mathbf{L},\mathbf{L}|t}\}$ changes across $t$. And an $\mathsf{T} \rightarrow \mathsf{X}_i^{(k)} \in \mathcal{G}^{aug}$ if and only if the estimated $\{\hat{\phi}_{i|t}^{(k)}\}$ changes across $t$.*

The invariant-anchor assumption in Assumption 7 (A1) is to resolve the scaling indeterminacy of the effective measurement coefficients; without at least one invariant anchor per latent variable, one cannot determine whether the latent variable itself changed or whether the change is due to an arbitrary rescaling. Compared to the generality of Phase 1, Phase 2 is narrower as it requires not only structure identifiability, but also parameter identifiability in latent-variable models. For one-factor models, change localization is possible as the parameters are identifiable up to only trivial indeterminacies (Bollen, 1989). For more general multi-factor models, however, parameters are typically identifiable only up to orthogonal transformations (Dong et al., 2024b), which makes change detection across domains challenging without further assumptions. That said, extending Phase 2 beyond one-factor models is still possible and we leave it for future work.

## 5. Experiments

### 5.1. Synthetic Setting, Baselines, and Evaluation Metric

We use synthetic data to validate the proposed tests and LCD-NOD. Specifically, to simulate i.i.d. data from nonstationary POLCMs, we randomly generate DAG where each $\mathsf{V}_i$ has a 0.5 possibility to be influenced by $\mathsf{T}$, i.e., $h_{:,i}$ and $g_i$ are a function of domain index $\mathsf{T} \in \{1, ..., 10\}$ and $\mathsf{T}$ is sampled from a categorical distribution with parameters $[0.25, 0.2, 0.15, 0.1, 0.1, 0.075, 0.05, 0.05, 0.025]$. For those that are not influenced by $\mathsf{T}$, the corresponding $h_{:,i}$ and $g_i$ does not change across domain. The independent noise terms $\delta$ and $\epsilon$ are sampled from Gaussian with variance sampled uniformly from $[0.01, 0.1]$ and $[0.1, 1]$ respectively. For $h_{j,i}$ and $g_i$, they are set as randomly

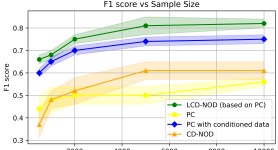
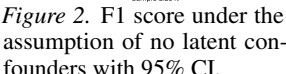

*Figure 2.* F1 score under the assumption of no latent confounders with 95% CI.

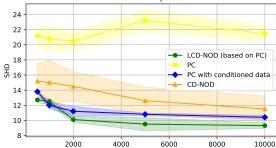

*Figure 3.* SHD under the assumption of no latent confounders with 95% CI.

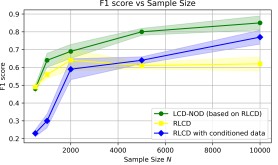

*Figure 4.* F1 under graphical assumptions by RLCD.

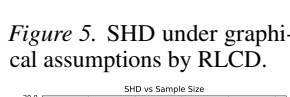

*Figure 5.* SHD under graphical assumptions by RLCD.

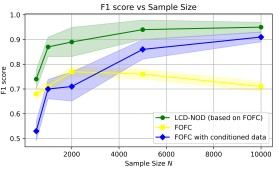

*Figure 6.* F1 regarding $\mathcal{G}$ of each method under the OFM.

*Figure 7.* SHD regarding $\mathcal{G}$ of each method under the OFM.

initialized polynomial functions parameterized by neural network. Thus, though $\epsilon_i$ is Gaussian, the effective noise $g_i(\mathsf{T}, \epsilon_i)$ is generally non-gaussian.

To validate the proposed conditional rank test, we employ the classical CCA-based rank test (Bartlett, 1947) on the unconditional covariance matrix, and the standard rank test with conditional covariance matrix with p value of a random domain and average p value across domains, as baselines. We refer to our proposed two tests as Proposed Conditional Rank Test and Proposed Permutation Conditional Rank Test. We compare the Type-I and Type-II errors of each method under both Gaussianity and non-Gaussianity and consider sample sizes of 300, 500, 800, 1000, 1500, 2000 with 10 random seeds to report the average performance.

To validate LCD-NOD, we employ three classical equality-constraint-based causal discovery methods, PC (Spirtes et al., 2000), FOFC (Kummerfeld & Ramsey, 2016), and RLCD (Dong et al., 2024a), and see whether the first phase can successfully upgrade these methods to handle the nonstationary scenario, in terms of F1 score (bigger better) and SHD (smaller better). These three methods consider structures without latent variables (Spirtes et al., 2000) (on average 20 observed variable), structures of one factor model (Silva et al., 2003) (on average 20 observed and 5 latent variables), and structures involving latent variables where all variables can be flexibly related (Dong et al., 2024a) (on average 20 observed and 5 latent variables).

The Phase 2 of LCD-NOD is to determine which variables are directly influenced by $\mathsf{T}$. Given that we allow the presence of latent variables, consider the whole structure, and allow $\mathsf{T}$ to influence both latent and observed variables, to our best knowledge, no existing method can achieve this result. Below is a comparison with the very related settings: CD-NOD must assume no latent variable, (Jaber et al., 2020) only cares structure among observed and only allows interventions on observed, and LIT (Yang et al., 2024) only allow changes on exogenous noises and can only find edges from $\mathsf{T}$ to $\mathsf{X}$. Thus, we mainly focus on comparing with the output of Phase 2 to ground truth. We focus on the end-2-end setting that take the data to LCD-NOD and get result of Phase 2 directly, as in Section 5.3, while also consider the disentangled setting detailed in Section B.3.

We also compare the end-to-end performance of LCD-NOD with (i) LIT (Yang et al., 2024) and (ii) UT-IGSP (Squires

et al., 2020). To accommodate the constraints of LIT and UT-IGSP, this comparison is conducted under specific conditions that they require: allowing changes only in exogenous noises and identifying edges only from $\mathsf{T}$ to $\mathsf{X}$. The result can be found in Sec. B.4, where LCD-NOD still consistently outperforms them though the setting is in favor of them.

### 5.2. Type-I and Type-II Control of the Proposed Tests

The results can be found in Figures 8 to 11. In words, both proposed conditional rank tests can properly control the Type-I errors (with significance level $\alpha = 0.05$). At the same time, the Type-II errors of our proposed tests are consistently smaller than the baselines, which illustrates the better test power of our proposed method. More detailed analysis can be found in Section B.1.

### 5.3. LCD-NOD Performance on Synthetic Data

In this section, we employ synthetic data to validate the effectiveness of LCD-NOD. For Phase 1 of LCD-NOD, we compare PC, FOFC, and RLCD with their versions upgraded with LCD-NOD. We also compare with the naive conditioning upgrade strategy, referred to as PC with conditioned data, FOFC with conditioned data, and RLCD with conditioned data, respectively. We also compare with CD-NOD, which also handles nonstationarity but requires the absence of latent variables.

Figures 2 and 3 give the performance of each method in cases without latent variables. As shown, LCD-NOD consistently surpasses baselines in terms of both F1 score and SHD, across different sample sizes. The performance under the one factor model assumption (Silva et al., 2003), and under the graphical assumption required by RLCD (Dong et al., 2024a) are given in Figures 6 and 7 and Figures 4 and 5, respectively. Similarly, LCD-NOD also outperforms all baselines, and the performance of LCD-NOD becomes better with the increase of sample size, while the original version of PC, FOFC, and RLCD do

not. This validates LCD-NOD as a general augmentation scheme of existing constraint-based causal discovery methods to handle the nonstationary scenario.

Last, we examine the end-2-end performance of Phase 2 of LCD-NOD. Specifically, we compare the output of Phase 2, i.e., the structure among $\mathbf{X}$, $\mathbf{L}$ and $\mathsf{T}$, with the ground truth.The results are in Figures 14 and 15, where the Phase 2 performs quite well in terms of recovering the augmented structure, and benefits from the increase of sample size. E.g., when sample size is 2000 per domain, Phase 2 achieves 0.83 F1 and 6.5 SHD and it improves to 0.85 F1 and 5.7 SHD when the sample size becomes 5000. These results empirically validate the effectiveness of LCD-NOD. We also found that in Phase 2 it is easier to detect changes in observed variables. An intuitive explanation is as follows. A change in an observed variable, typically amounts to only a change in its measurement process, characterized by one set of conditional coefficients to estimate. However, a change in a latent variable can involve multiple other latent variables in the underlying structure, and hence multiple sets of coefficients conditioning on different latent variables need to be estimated. This is also aligned with the reason why we need to explicitly impose Asm. 7 A2 and to address the indeterminacies of latent variables to make such detection feasible.

### 5.4. LCD-NOD Performance on Real-world Data

We employ Big Five personality dataset (`openpsychometrics.org`) to show the real-life applicability of LCD-NOD. By taking country as the domain index, we found that causal mechanisms among certain dimensions, e.g., agreeableness and extroversion, indeed exhibits nonstationarity, which aligns with psychometric studies. Please kindly refer to Section B.2.

## 6. Related Work

**Causal discovery with latent variables:** Early works that accommodate latent confounders were the FCI method (Spirtes et al., 2000; Richardson & Spirtes, 2002; Zhang, 2008), which utilize conditional independence tests to identify causal relations among observed variables while accounting for latent confounders. While FCI does not make any assumption about the latent structure, it often produces less informative outputs, e.g., provides no information about relationships among latent variables. On the other hand, FCI has shown to be maximally informative when considering nonparametric conditional independence constraints. Therefore, to go beyond them, further constraints or information have been leveraged, such as those relying on parametric assumptions, as discussed in Section 1. A common approach is to use rank constraints (Sullivant et al., 2010), a generalization of the classical Tetrad constraints (Spirtes et al., 2000) and conditional independence constraints. This leads to a number of works that rely on different graphical

assumptions (Silva et al., 2003; Huang et al., 2022; Dong et al., 2024a; Ng et al., 2024; Dong et al., 2026), with selection bias (Dai et al., 2025) or data discretization (Dong et al., 2025; Sun et al., 2025). Alternatively, several other methods also rely on higher-order information (Shimizu et al., 2009; Cai et al., 2019; Salehkaleybar et al., 2020; Xie et al., 2020; Adams et al., 2021).

**Causal discovery from heterogeneous data:** Score-based methods have been developed to infer structure from heterogeneous data, which include those based on greedy search (Hauser & Bühlmann, 2012; Squires et al., 2020) or continuous optimization (Brouillard et al., 2020). On the other hand, Huang et al. (2020) developed a constraint-based method that relies on conditional independence test, while Mooij et al. (2020) proposed a general framework that can incorporate different causal discovery methods.

**Causal discovery with latent variables and causal representation learning from heterogenous data:** Magliacane et al. (2016); Kocaoglu et al. (2019) proposed constraint-based methods that rely on conditional independence tests to recover ancestral structures over the observed variables, similar in spirit to FCI. In contrast, a related line of work, causal representation learning (Schölkopf et al., 2021), aims to infer both the latent causal variables and the causal structure among them. A special case of causal representation learning is nonlinear ICA which assumes that the latent variables are independent (Hyvarinen & Morioka, 2017; 2016; Hyvarinen et al., 2019; Hyvärinen et al., 2023). These methods also leverage interventional or heterogeneous data, such as single-node interventions (Ahuja et al., 2023; Squires et al., 2023; von Kügelgen et al., 2023; Zhang et al., 2023; Varıcı et al., 2024a) or multi-node interventions (Jin & Syrgkanis, 2023; Zhang et al., 2024; Varıcı et al., 2024b; Bing et al., 2024; Ng et al., 2025). Furthermore, some of them require hard interventions, such as von Kügelgen et al. (2023); Bing et al. (2024). Note that this line of approaches based on causal representation learning typically make certain assumptions: (1) no causal edges exist among observed variables or from observed to latent variables, and (2) the generative process from latent variables is deterministic (except several works including Khemakhem et al. (2020); Lachapelle et al. (2024)) and invariant across domains. In our work, we consider a more general setting that relaxes these assumption.

## 7. Conclusion

This work formulates a class of nonstationary models, shows the constraints, and develops principled tests and latent variable causal discovery method under nonstationarity,

## Impact Statement

This work is to advance the field of Machine Learning. There are many potential societal consequences of our work, none which we feel must be specifically highlighted here.

## Acknowledgements

We would like to acknowledge the support from NSF Award No. 2229881, AI Institute for Societal Decision Making (AI-SDM), the National Institutes of Health (NIH) under Contract R01HL159805, and grants from Quris AI, Florin Court Capital, MBZUAI-WIS Joint Program, and the Al Deira Causal Education project.

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

# Appendix

# A. Proofs

## A.1. Proof of Theorem 1

**Theorem 1** (Equivalence Relation between $\Theta(\mathcal{G})$ under POLCMs and $\Phi(\mathcal{G})$ under Nonstationary POLCMs). *Given a DAG $\mathcal{G}$, let $\Theta(\mathcal{G})$ be the observational covariance set of $\mathcal{G}$ under POLCMS, as in Definition 3, and let $\Phi(\mathcal{G})$ be the observational conditional covariance set of $\mathcal{G}$ under Nonstationary POLCMs, as in Definition 4. We have that $\Theta(\mathcal{G}) = \Phi(\mathcal{G})$.*

*Proof.* We first show the following Theorem 9.

**Lemma 9** (Lemma for proof of Theorem 1).

$$\mathbb{E}_{p(\mathsf{V}|\mathsf{T})}[(I - H(\mathsf{T})^\top)^{-1}g(\mathsf{T})g(\mathsf{T})^\top(I - H(\mathsf{T})^\top)^{-\top}] \tag{8}$$

$$=\mathbb{E}_{p(\delta,\epsilon|\mathsf{T})}[(I - H(\mathsf{T})^\top)^{-1}g(\mathsf{T})g(\mathsf{T})^\top(I - H(\mathsf{T})^\top)^{-\top}] \tag{9}$$

$$=(I - \mathbb{E}_{p(\delta,\epsilon|\mathsf{T})}H(\mathsf{T})^\top)^{-1}\mathbb{E}_{p(\delta,\epsilon|\mathsf{T})}[g(\mathsf{T})g(\mathsf{T})^\top](I - \mathbb{E}_{p(\delta,\epsilon|\mathsf{T})}H(\mathsf{T})^\top)^{-\top}. \tag{10}$$

*Proof of Lemma 9.* Let $\Omega_g = \mathbb{E}_{p(\delta,\epsilon|\mathsf{T})}[gg^\top]$. As $g_i$ are mutually independent given $\mathsf{T}$, $\Omega_g$ is a diagonal matrix. As $g$ and $H$ are independent given $\mathsf{T}$, we have: $\mathbb{E}_{p(\delta,\epsilon|\mathsf{T})}[(I - H^\top)^{-1}gg^\top(I - H^\top)^{-\top}] = \mathbb{E}_{p(\delta,\epsilon|\mathsf{T})}[(I - H^\top)^{-1}\Omega_g(I - H^\top)^{-\top}]$. Now consider the i-th row and j-th column of it, i.e., $\mathbb{E}_{p(\delta,\epsilon|\mathsf{T})}[(I - H^\top)^{-1}\Omega_g(I - H^\top)^{-\top}]_{[i,j]} = \mathbb{E}_{p(\delta,\epsilon|\mathsf{T})}[(I - H^\top)^{-1}\Omega_g(I - H^\top)^{-\top}]_{[i,j]} = \mathbb{E}_{p(\delta,\epsilon|\mathsf{T})}[\sum_{P_1,P_2\in\mathcal{T}(\mathsf{V}_i,\mathsf{V}_j)}\Omega_{g\,\text{top}(P_1,P_2)}H^{P_1}H^{P_2}]$, where $H^P = \Pi_{j\to i\in P}h_{j,i}(\mathsf{T},\delta_{j,i})$, $\mathcal{T}(\mathsf{V}_i,\mathsf{V}_j)$ refers to the set of treks between $\mathsf{V}_i$ and $\mathsf{V}_j$, and $\text{top}(P_1,P_2)$ is the common source of the trek $(P_1,P_2)$. As $P_1, P_2$ do not share edges, we have

$$\mathbb{E}_{p(\delta,\epsilon|\mathsf{T})}\Big[\sum_{P_1,P_2\in\mathcal{T}(\mathsf{V}_i,\mathsf{V}_j)}\Omega_{g\,\text{top}(P_1,P_2)}H^{P_1}H^{P_2}\Big] \tag{11}$$

$$= \sum_{P_1,P_2\in\mathcal{T}(\mathsf{V}_i,\mathsf{V}_j)}\Omega_{g\,\text{top}(P_1,P_2)}\mathbb{E}_{p(\delta,\epsilon|\mathsf{T})}[H]^{P_1}\mathbb{E}_{p(\delta,\epsilon|\mathsf{T})}[H]^{P_2} \tag{12}$$

$$= (I - \mathbb{E}_{p(\delta,\epsilon|\mathsf{T})}[H]^\top)^{-1}\Omega_g(I - \mathbb{E}_{p(\delta,\epsilon|\mathsf{T})}[H]^\top)^{-\top}. \tag{13}$$

$\square$

Note that the above proof does not treat random coefficients as deterministic. It only uses that all h are mutually independent given $\mathsf{T}$. The detailed derivation is:

$$\mathbb{E}[\sum\Omega_{g\text{top}(P_1,P_2)}H^{P_1}H^{P_2}|\mathsf{T}] = \sum\Omega_{g\text{top}(P_1,P_2)}\mathbb{E}[H^{P_1}H^{P_2}|\mathsf{T}].$$

As the structure is acyclic, there is no repeated edges in trek $P_1P_2$. Thus $H^{P_1}H^{P_2}$ is a multiplication of independent terms given $\mathsf{T}$ and thus

$$\mathbb{E}[\sum\Omega_{g\text{top}(P_1,P_2)}H^{P_1}H^{P_2}|\mathsf{T}] = \sum\Omega_{g\text{top}(P_1,P_2)}\mathbb{E}[H^{P_1}H^{P_2}|\mathsf{T}] = \sum\Omega_{g\text{top}(P_1,P_2)}\mathbb{E}[H|\mathsf{T}]^{P_1}\mathbb{E}[H|\mathsf{T}]^{P_2}.$$

Plus, in the integration form of the expectation, the trek expansion is applied pointwise to each realization of $H$. Thus the trek decomposition itself is purely algebraic, exactly as in the deterministic-coefficient case.

Now, given Lemma 9, we prove Theorem 1.

(i) For every element of $\Theta(\mathcal{G})$ generated by $(F, \Omega)$, let $\mathbb{E}_{p(\delta,\epsilon|\mathsf{T})}[H] = F$ and $\mathbb{E}_{p(\delta,\epsilon|\mathsf{T})}[gg^\top] = \Omega$. Then $\Phi(\mathcal{G})$ can generate the same element. (ii) For every element of $\Phi(\mathcal{G})$ generated by $(H, g)$, let $F = \mathbb{E}_{p(\delta,\epsilon|\mathsf{T})}[H]$ and $\Omega = \mathbb{E}_{p(\delta,\epsilon|\mathsf{T})}[gg^\top]$. Then $\Theta(\mathcal{G})$ can generate the same element. Taking these two together, we have $\Theta(\mathcal{G}) = \Phi(\mathcal{G})$. $\square$

### A.2. Proof of Theorem 2

**Corollary 10** (Equality and Inequality Constraints Under Nonstationary). *If $\mathcal{G}$ implies an equality or inequality constraint on $\Theta(\mathcal{G})$, then $\mathcal{G}$ also implies the same constraint on $\Phi(\mathcal{G})$, and vice versa.*

*Proof.* If $\mathcal{G}$ implies a constraint on $\Theta(\mathcal{G})$, then by $\Theta(\mathcal{G}) = \Phi(\mathcal{G})$ as in Theorem 1, $\mathcal{G}$ also implies the same constraint on $\Phi(\mathcal{G})$, and vice versa. $\qquad\square$

### A.3. Proof of Theorem 3

**Corollary 11** (Rank and T-separation for Nonstationary POLCMs). *Given two sets of variables $\mathbf{A}$ and $\mathbf{B}$ generated by Nonstationary POLCMs with DAG $\mathcal{G}$ and assume rank faithfulness (Spirtes, 2013; Dong et al., 2024a). We have:*

$$rank(\Sigma_{\mathbf{A},\mathbf{B}|\mathsf{T}}) = \min\{|\mathbf{C_A}| + |\mathbf{C_B}| : (\mathbf{C_A}, \mathbf{C_B})$$
$$t\text{-separates}\,\mathbf{A}\text{ from }\mathbf{B}\text{ in }\mathcal{G}\}, \tag{5}$$

*where $\Sigma_{\mathbf{A},\mathbf{B}|\mathsf{T}}$ is the cross-covariance over $\mathbf{A}$ and $\mathbf{B}$ conditional on $\mathsf{T}$.*

*Proof.* By the relation between rank and t-separation in (Sullivant et al., 2010) for stationary linear causal models, we have that $rank(\Sigma_{\mathbf{A},\mathbf{B}|\mathsf{T}}) = \min\{|\mathbf{C_A}| + |\mathbf{C_B}| : (\mathbf{C_A}, \mathbf{C_B})$ t-separates $\mathbf{A}$ from $\mathbf{B}$ in $\mathcal{G}\}$. As rank constraints are a subset of equality constraints entailed by $\mathcal{G}$ for stationary linear causal models, we have $rank(\Sigma_{\mathbf{A},\mathbf{B}})$ generated by Definition 2 equals $rank(\Sigma_{\mathbf{A},\mathbf{B}|\mathsf{T}})$ generated by Definition 1. Thus, we have $rank(\Sigma_{\mathbf{A},\mathbf{B}|\mathsf{T}}) = \min\{|\mathbf{C_A}| + |\mathbf{C_B}| : (\mathbf{C_A}, \mathbf{C_B})$ t-separates $\mathbf{A}$ from $\mathbf{B}$ in $\mathcal{G}\}$ for nonstationary POLCMs. $\qquad\square$

### A.4. Proof of Theorem 4

**Theorem 4** (Likelihood Ratio Statistics for Rank of Conditional Covariance). *Given two sets of random variables $\mathbf{A}$ and $\mathbf{B}$ with $|\mathbf{A}| = P, |\mathbf{B}| = Q$, $K = \min(P, Q)$ and $\mathbf{A} \cup \mathbf{B}$ are jointly gaussian given $\mathsf{T}$. Assume the null hypothesis $\mathcal{H}_0$ is $rank(\Sigma_{\mathbf{A},\mathbf{B}|\mathsf{T}}) \leq k$, the likelihood ratio statistics $\Lambda(k)$ is*

$$\sum_{t \in supp(\mathsf{T})} \frac{P + Q + 1 - 2N_t}{2} \ln\left( \prod_{i=k+1}^{K} (1 - r_{t,i}^2) \right), \tag{6}$$

*where $N_t$ is the number of data points such that $\mathsf{T} = t$, and $r_{t,i}$ is the $i$-th canonical correlation between $\mathbf{A}$ and $\mathbf{B}$ conditioned on $\mathsf{T} = t$. Further, $\Lambda(k)$ converges in distribution to $\chi_{df}^2$, with degree of freedom $\sum_{t \in supp(\mathsf{T})} (P - k)(Q - k)$.*

*Proof.* Let $D_{\mathbf{X}}$ be the observed data of $\mathbf{X} = \mathbf{A} \cup \mathbf{B}$ and $D_{\mathbf{X}}^t$ be the observed data of $\mathbf{X}$ conditioned on $\mathsf{T} = t$. The log-likelihood ratio statistics is:

$$\Lambda(k) = 2 \log \frac{\sup_{\Sigma_{\mathbf{A},\mathbf{B}|\mathsf{T}} \in \Theta_0} L(D_{\mathbf{X}}; \Sigma_{\mathbf{X}})}{\sup_{\Sigma_{\mathbf{A},\mathbf{B}|\mathsf{T}} \in \Theta} L(D_{\mathbf{X}}; \Sigma_{\mathbf{X}})} \tag{14}$$

$$= -2 \log \frac{\sup_{\Sigma_{\mathbf{A},\mathbf{B}|\mathsf{T}} \in \Theta_0} \Pi_{t \in supp(\mathsf{T})} P(D_{\mathbf{X}}^t | \mathsf{T} = t; \Sigma_{\mathbf{X}|\mathsf{T}=t}) P(\mathsf{T} = t)}{\sup_{\Sigma_{\mathbf{A},\mathbf{B}|\mathsf{T}} \in \Theta} \Pi_{t \in supp(\mathsf{T})} P(D_{\mathbf{X}}^t | \mathsf{T} = t; \Sigma_{\mathbf{X}|\mathsf{T}=t}) P(\mathsf{T} = t)} \tag{15}$$

$$= -2 \log \frac{\sup_{\Sigma_{\mathbf{A},\mathbf{B}|\mathsf{T}} \in \Theta_0} \Pi_{t \in supp(\mathsf{T})} P(D_{\mathbf{X}}^t | \mathsf{T} = t; \Sigma_{\mathbf{X}|\mathsf{T}=t})}{\sup_{\Sigma_{\mathbf{A},\mathbf{B}|\mathsf{T}} \in \Theta} \Pi_{t \in supp(\mathsf{T})} P(D_{\mathbf{X}}^t | \mathsf{T} = t; \Sigma_{\mathbf{X}|\mathsf{T}=t})} \tag{16}$$

$$= \sum_{t \in supp(T)} -2 \log \frac{\sup_{\Sigma_{\mathbf{A},\mathbf{B}|\mathsf{T}} \in \Theta_0} P(D_{\mathbf{X}}^t | \mathsf{T} = t; \Sigma_{\mathbf{X}|\mathsf{T}=t})}{\sup_{\Sigma_{\mathbf{A},\mathbf{B}|\mathsf{T}} \in \Theta} P(D_{\mathbf{X}}^t | \mathsf{T} = t; \Sigma_{\mathbf{X}|\mathsf{T}=t})}. \tag{17}$$

For each $t$, the likelihood ratio statistics conditioned on $t$ is $\lambda(k, t) = \frac{\sup_{\Sigma_{\mathbf{A},\mathbf{B}|\mathsf{T}} \in \Theta_0} P(D_{\mathbf{X}}^t | \mathsf{T}=t; \Sigma_{\mathbf{X}|\mathsf{T}=t})}{\sup_{\Sigma_{\mathbf{A},\mathbf{B}|\mathsf{T}} \in \Theta} P(D_{\mathbf{X}}^t | \mathsf{T}=t; \Sigma_{\mathbf{X}|\mathsf{T}=t})}$. The numerator $\sup_{\Sigma_{\mathbf{A},\mathbf{B}|\mathsf{T}} \in \Theta_0} P(D_{\mathbf{X}}^t | \mathsf{T} = t; \Sigma_{\mathbf{X}|\mathsf{T}=t})$ is a problem of MLE under rank constraint. By (Bartlett, 1947; Muirhead, 2009), the

problem can be solved by calculating the empirical canonical correlation problem between the observations of $\mathbf{A}, \mathbf{B}$. More specifically, we have that $\lambda(k, t) = -\left(N_t - \frac{P+Q+1}{2}\right) \ln(\Pi_{i=k+1}^{\min(P,Q)}(1 - r_{t,i}^2))$, and thus Equation (6).

Now we show the asymptotic distribution of the statistics in Equation (6). As $\lambda(k, t)$ converges in distribution to $\chi_{(P-k)(Q-k)}^2$, and for different $t$, $\lambda(k, t)$ are mutually independent, by the use of continuous mapping theorem, $\Lambda(k) = \sum_{t \in \text{supp}(\mathsf{T})} \lambda(k, t)$ converges in distribution to $\chi_{\sum_{t \in \text{supp}(\mathsf{T})}(P-k)(Q-k)}^2$. $\qquad \square$

### A.5. Proof of Theorem 5

**Theorem 5** (Permutation-based Rank Test of Conditional Covariance). *Given random vectors $\mathbf{A}$ and $\mathbf{B}$ with $|\mathbf{A}| = P, |\mathbf{B}| = Q$ and their corresponding data matrices $\boldsymbol{D}^{\mathbf{A}|t} \in \mathbb{R}^{N_t \times P}$ and $\boldsymbol{D}^{\mathbf{B}|t} \in \mathbb{R}^{N_t \times Q}$ for all $t \in \text{supp}(\mathsf{T})$ generated following Definition 1, and the null hypothesis $\mathcal{H}_0$ as $\text{rank}(\Sigma_{\mathbf{A},\mathbf{B}|\mathsf{T}}) \leq k$, the following test is valid.*

*(i) For each $t \in \text{supp}(\mathsf{T})$, solve the following CCA problem by SVD: $\boldsymbol{U}\boldsymbol{S}\boldsymbol{V} = \hat{\Sigma}_{\mathbf{A}|t}^{-\frac{1}{2}}\hat{\Sigma}_{\mathbf{A},\mathbf{B}|t}\hat{\Sigma}_{\mathbf{B}|t}^{-\frac{1}{2}}$, $\boldsymbol{R}^{\mathbf{A}} = \hat{\Sigma}_{\mathbf{A}|t}^{-\frac{1}{2}\top}\boldsymbol{U}$, and $\boldsymbol{R}^{\mathbf{B}} = \hat{\Sigma}_{\mathbf{B}|t}^{-\frac{1}{2}\top}\boldsymbol{V}^{\top}$, and define canonical vectors $\mathbf{C}_{\mathbf{A}|t} = \boldsymbol{R}^{\mathbf{A}\top}\mathbf{A}|t$ and $\mathbf{C}_{\mathbf{B}|t} = \boldsymbol{R}^{\mathbf{B}\top}\mathbf{B}|t$, with data matrices as $\boldsymbol{D}^{\mathbf{C}_{\mathbf{A}}|t} = \boldsymbol{D}^{\mathbf{A}|t}\boldsymbol{R}^{\mathbf{A}}$ and $\boldsymbol{D}^{\mathbf{C}_{\mathbf{B}}|t} = \boldsymbol{D}^{\mathbf{B}|t}\boldsymbol{R}^{\mathbf{B}}$, respectively.*

*(ii) Define a set of random permutation matrices $\mathcal{P} = \{\boldsymbol{P}_t : t \in \text{supp}(\mathsf{T}), \boldsymbol{P}_t \text{ is a } N_t \times N_t \text{ permutation matrix}\}$.*

*(iii) Define the statistics under each $\mathcal{P}$, $\Lambda(k, \mathcal{P})$, as*

$$\sum_{t \in \text{supp}(\mathsf{T})} \frac{P + Q + 1 - 2N_t}{2} \ln\left(\prod_{i=1}^{K-k}(1 - \hat{r}_{t,\boldsymbol{P}_t,i}^2)\right) \tag{7}$$

*where $\hat{r}_{t,\boldsymbol{P}_t,i}^2$ refers to the $i$-th CCA score between data matrices $\boldsymbol{P}_t \boldsymbol{D}_{:,k:}^{\mathbf{C}_{\mathbf{A}}|t}$ and $\boldsymbol{D}_{:,k:}^{\mathbf{C}_{\mathbf{B}}|t}$.*

*(iv) Calculate the p-value as $p_k = \mathbb{E} \, \mathbf{1}_{[\Lambda(k,\mathcal{P}) \geq \Lambda(k,\mathcal{I})]}$, where the expectation is taken over random $\mathcal{P}$ and $\mathcal{I} = \{\boldsymbol{I}_t : t \in \text{supp}(\mathsf{T}), \boldsymbol{I}_t \text{ is the } N_t \times N_t \text{ identity matrix}\}$.*

*Proof.* We need to show that, each $\Lambda(k, \mathcal{P})$ is a resampling from the distribution of $\Lambda(k, \mathcal{I})$.

We first look at each term of $\Lambda(k, \mathcal{I})$, i.e., $\frac{P+Q+1-2N_t}{2} \ln(\prod_{i=1}^{K-k}(1 - \hat{r}_{t,\boldsymbol{I}_t,i}^2))$, which is a function of $\boldsymbol{D}^{\mathbf{A}|t}$ and $\boldsymbol{D}^{\mathbf{B}|t}$, which is produced by conditioning on $\mathsf{T} = t$. Thus there is no overlapping data points used across $\frac{P+Q+1-2N_t}{2} \ln(\prod_{i=1}^{K-k}(1 - \hat{r}_{t,\boldsymbol{I}_t,i}^2))$ for different $t$. As all data points are i.i.d. samples, we have that for all $t$, $\frac{P+Q+1-2N_t}{2} \ln(\prod_{i=1}^{K-k}(1 - \hat{r}_{t,\boldsymbol{I}_t,i}^2))$ are mutually independent. Which means we can resample $\frac{P+Q+1-2N_t}{2} \ln(\prod_{i=1}^{K-k}(1 - \hat{r}_{t,\boldsymbol{I}_t,i}^2))$ for each $t$ in order to resample $\Lambda(k, \mathcal{I})$.

Now we show that we can resample $\frac{P+Q+1-2N_t}{2} \ln(\prod_{i=1}^{K-k}(1 - \hat{r}_{t,\boldsymbol{I}_t,i}^2))$ for each $t$ by permutation. The core is to show the asymptotically exchangeability of random vectors $(\mathbf{C}_{\mathbf{A}|t})_{k:}$ and $(\mathbf{C}_{\mathbf{B}|t})_{k:}$ under the null hypothesis. The proof techniques of this part are inspired by Dong et al. (2025). Asymptotically $\hat{\Sigma}_{\mathbf{A}|t}$, $\hat{\Sigma}_{\mathbf{B}|t}$, and $\hat{\Sigma}_{\mathbf{A},\mathbf{B}|t}$ converge in probability to $\Sigma_{\mathbf{A}|t}$, $\Sigma_{\mathbf{B}|t}$, and $\Sigma_{\mathbf{A},\mathbf{B}|t}$, respectively. Following the continuity and uniqueness of SVD shown in Dong et al. (2025); Kato (2013); Kunisky; Bochnak et al. (2013), we know that $\boldsymbol{R}^{\mathbf{A}}$ and $\boldsymbol{R}^{\mathbf{B}}$ also converge in probability to their population counterpart. Thus, under the null hypothesis that $\text{rank}(\Sigma_{\mathbf{A},\mathbf{B}|\mathsf{T}}) \leq k$, asymptotically the population covariance between $(\mathbf{C}_{\mathbf{A}|t})_{k:}$ and $(\mathbf{C}_{\mathbf{A}|t})_{k:}$ are all zeros. As all variables are generated following Definition 1, given $\mathsf{T}$, all variables can be considered as generated following a stationary linear causal model with the same graph; By Theorem 1, we have that the conditional covariance set is the same as a stationary linear causal model. Under the rank faithfulness assumption, we have that SVD rotation matrices do not accidentally drop rank (or the set of parameters that induces drop of rank has measure zero). Here the rank faithfulness is used only to rule out degenerate cancellations, so these zero cross-covariances reflect genuine absence of shared exogenous sources in the two rotated trailing coordinates. Therefore, given that the cross-covariance of $(\mathbf{C}_{\mathbf{A}|t})_{k:}$ and $(\mathbf{C}_{\mathbf{A}|t})_{k:}$ are all zeros, they do not share common variable / noise terms - the two rotated trailing coordinates are functions of disjoint sets of independent noises and are thus asymptotically independent within each domain. Thus, $(\mathbf{C}_{\mathbf{A}|t})_{k:}$ and $(\mathbf{C}_{\mathbf{A}|t})_{k:}$ are asymptotically independent, and thus we establish the asymptotic exchangeability.

Note that the data matrices for $(\mathbf{C}_{\mathbf{A}|t})_{k:}$ and $(\mathbf{C}_{\mathbf{A}|t})_{k:}$ are $\boldsymbol{D}_{:,k:}^{\mathbf{C}_{\mathbf{A}}|t}$ and $\boldsymbol{D}_{:,k:}^{\mathbf{C}_{\mathbf{B}}|t}$ respectively. By the exchangeability, asymptotically $P((\mathbf{C}_{\mathbf{A}|t})_{k:}, (\mathbf{C}_{\mathbf{B}|t})_{k:}) = P((\mathbf{C}_{\mathbf{A}|t})_{k:})P((\mathbf{C}_{\mathbf{B}|t})_{k:})$, and thus we can permute one of their data matrix in order to resample from their joint distribution, as $\boldsymbol{P}_t \boldsymbol{D}_{:,k:}^{\mathbf{C}_{\mathbf{A}}|t}$ and $\boldsymbol{D}_{:,k:}^{\mathbf{C}_{\mathbf{B}}|t}$.

Therefore, $\frac{P+Q+1-2N_t}{2}\ln(\prod_{i=1}^{K-k}(1-\hat{r}^2_{t,\boldsymbol{I}_t,i}))$ can be resampled by $\frac{P+Q+1-2N_t}{2}\ln(\prod_{i=1}^{K-k}(1-\hat{r}^2_{t,\boldsymbol{P}_t,i}))$, and thus $\Lambda(k,\mathcal{I})$ can be resampled by $\Lambda(k,\mathcal{P})$. Therefore, the empirical distribution of $\Lambda(k,\mathcal{I})$ can be estimated and thus asymptotically the p-value by $p_k = \mathbb{E}\,\mathbf{1}_{[\Lambda(k,\mathcal{P})\geq\Lambda(k,\mathcal{I})]}$ follows a uniform distribution. $\qquad\square$

## A.6. Proof of Theorem 6

**Theorem 6** (Structure Identifiability of LCD-NOD Phase 1). *Given an equality-constraint-based causal discovery method $\mathcal{M}$, if $\mathcal{M}$ asymptotically identifies $\mathcal{G}$ up to equivalence class $\mathcal{C}$ under POLCMs and graphical assumption $\mathcal{A}$, then the Phase 1 of LCD-NOD, i.e., the augmented $\mathcal{M}$, asymptotically identifies $\mathcal{G}$ up to $\mathcal{C}$, under Nonstationary POLCMs and $\mathcal{A}$.*

*Proof.* In the large sample limit, the input to $\mathcal{M}$ and the augmented $\mathcal{M}$ are the population covariance over $\mathbf{X}$ and the conditional population covariance over $\mathbf{X}$ respectively. In other words, the inputs are an element of $\Theta(\mathcal{G})$ and an element of $\Phi(\mathcal{G})$ respectively. Under faithfulness, these two elements contain the same set of constraints as $\Theta(\mathcal{G})$ and $\Phi(\mathcal{G})$ respectively. By Theorem 2, these two elements contain the same set of equality constraints. Further, as $\mathcal{M}$ and the augmented $\mathcal{M}$ only makes use of equality constraints, the two algorithms will have exactly the same output, and thus the augmented $\mathcal{M}$ also asymptotically identifies $\mathcal{G}$ up to $\mathcal{C}$. $\qquad\square$

## A.7. Proof of Theorem 8

**Theorem 8** (Identification of changing variables). *Suppose measurements $\mathbf{X}$ are generated following Definition 6 and let $\{\hat{\Sigma}_{\mathbf{L},\mathbf{L}|t}, \hat{\omega}_t^{(1)}, \hat{\omega}_t^{(2)}, \hat{\phi}_t^{(1)}, \hat{\phi}_t^{(2)}\}$ be the model parameters estimated in each domain $\mathsf{T}=t$. Denote the normalized latent covariance matrices by $\{\hat{\Sigma}'_{\mathbf{L},\mathbf{L}|t} := \text{diag}(\hat{\omega}_t^{(1)})\,\hat{\Sigma}_{\mathbf{L},\mathbf{L}|t}\,\text{diag}(\hat{\omega}_t^{(1)})\}$. Then, under (A1) and (A2), $\mathsf{T}\to \mathsf{L}_i \in \mathcal{G}^{aug}$ if and only if for all subsets $\mathbf{L}_C \subseteq \mathbf{L} \setminus \{\mathsf{L}_i\}$, the conditional variances $\{\hat{\text{Var}}'_t(\mathsf{L}_i|\mathbf{L}_C)\}$ calculated from $\{\hat{\Sigma}'_{\mathbf{L},\mathbf{L}|t}\}$ changes across $t$. And an $\mathsf{T}\to\mathsf{X}_i^{(k)} \in \mathcal{G}^{aug}$ if and only if the estimated $\{\hat{\phi}_{i|t}^{(k)}\}$ changes across $t$.*

*Proof.* First, we show that using the indirect measurements from a single domain, the correspondences from latent variables to measurements can be determined, and the causal structure $\mathcal{G}$ among latent variables can be identified to its CPDAG. This is established in Theorem 2 and (Silva et al., 2003):

Suppose data is generated by the nonstationary one-factor model as in Definition 6, and assume rank faithfulness. The measurement clusters can first be identified, in that the cross-covariance matrix between two measured variables and all the remaining measured variables has a rank $< 2$ if and only if these two measured variables serve as measurements for a same latent variable. The CI relations among latent variables can then be identified with these clusters, in that any CI relation $\mathbf{L}_A \perp\!\!\!\perp \mathbf{L}_B | \mathbf{L}_C$ holds, if and only if the cross-covariance matrix between $\mathbf{X}_A \cup \mathbf{X}_C^{(1)}$ and $\mathbf{X}_B \cup \mathbf{X}_C^{(2)}$ is $|C|$, instead of higher. Here $\mathbf{X}_C^{(1)}$ and $\mathbf{X}_C^{(2)}$ are disjoint partitions of $\mathbf{X}_C$ such that $|\mathbf{X}_C^{(1)}|, |\mathbf{X}_C^{(2)}| \geq 2$. Then, with these CI relations recovered through ranks, one can apply e.g., PC as if having direct access to $\mathbf{L}$.

Then, we show that in additional to the model structure, it is also possible to consistently estimate (up to trivial indeterminacies) the model parameters from data in each domain. This allows us to estimate the conditional distributions $p(\mathsf{L}_i \mid \mathbf{L}_C, \mathsf{T}=t)$ directly, enabling the detection of changes:

Suppose we have access to measurements $\mathbf{X}$ in a single domain $\mathsf{T}=t$ generated by Definition 6. For simplicity, we drop the conditioning notations $(|\mathsf{T}=t)$ when the scope is clear. There are following unknown model parameters: $\Sigma_{\mathbf{L},\mathbf{L}} \in \mathbb{R}^{m\times m}$ and $\mu_{\mathbf{L}} \in \mathbb{R}^m$, the variance-covariance matrix and the mean vector among $\mathbf{L}$; $\omega^{(1)}, \omega^{(2)}, \mu^{(1)}, \mu^{(2)}, \phi^{(1)}, \phi^{(2)} \in \mathbb{R}^m$, the equivalent linear coefficients, intercepts, and exogenous noise variances of each latent variable's two pure measuring processes, where for $k=1,2$, $\omega_i^{(k)} = \mathbb{E}[h'_{i,k}(t,\delta'_{i,k})]$, $\mu_i^{(k)} = \mathbb{E}[g'_{i,k}(t,\epsilon'_{i,k})]$, and $\phi_i^{(k)} = \text{Var}[g'_{i,k}(t,\epsilon'_{i,k})]$.

To identify the model parameters, we first assume, without loss of generality, that each latent variable is dependent on at least one other latent variable. This assumption is necessary as otherwise its variance measurement parameters can be arbitrary. This assumption is also testable as isolated variables can be directly identified using marginal pairwise independence tests.

We further notice the trivial scaling and shifting indeterminacies: a latent variable can always be rescaled and shifted as long as its corresponding measurements are adjusted accordingly. However, we show that these are the only indeterminacies: Let $\hat{\Sigma}_{\mathbf{L},\mathbf{L}}, \hat{\mu}_{\mathbf{L}}, \hat{\omega}^{(1)}, \hat{\omega}^{(2)}, \hat{\mu}^{(1)}, \hat{\mu}^{(2)}, \hat{\phi}^{(1)}, \hat{\phi}^{(2)}$ be another set of model parameters (or estimators) that yield the same mean and covariance for $\mathbf{X}$. Let $\hat{\Sigma}_{\mathbf{L},\mathbf{L}}$ has unit diagonal, $\hat{\mu}_{\mathbf{L}} = 0$, and all entries of $\hat{\omega}^{(1)}$ are positive. Then, under these constraints, all

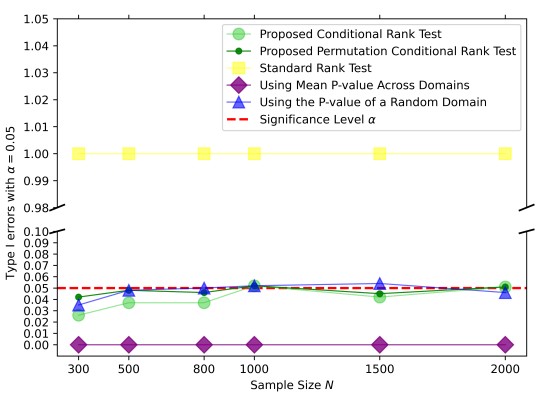

*Figure 8.* The probability of Type I errors with significance level $\alpha = 0.05$.

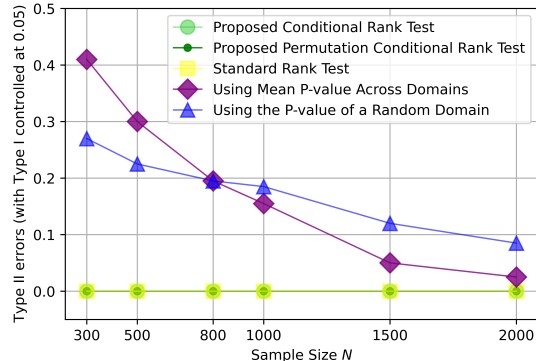

*Figure 9.* Type II errors (effective Type I controlled at 0.05).

remaining parameters are uniquely determined and can be expressed in closed form with the mean and covariance of $\mathbf{X}$. Specifically, they are:

- Off-diagonal entries of latent covariances: $\hat{\Sigma}_{\mathsf{L}_i, \mathsf{L}_j} = \text{sign}(\Sigma_{\mathsf{X}_i^{(1)}, \mathsf{X}_j^{(1)}}) \sqrt{\dfrac{\Sigma_{\mathsf{X}_i^{(1)}, \mathsf{X}_j^{(1)}} \Sigma_{\mathsf{X}_i^{(2)}, \mathsf{X}_j^{(2)}}}{\Sigma_{\mathsf{X}_i^{(1)}, \mathsf{X}_i^{(2)}} \Sigma_{\mathsf{X}_j^{(1)}, \mathsf{X}_j^{(2)}}}}$;

- Measuring weights: $\hat{\omega}_i^{(1)} = \sqrt{\dfrac{\Sigma_{\mathsf{X}_i^{(1)}, \mathsf{X}_i^{(2)}} \Sigma_{\mathsf{X}_i^{(1)}, \mathsf{X}_j^{(1)}}}{\Sigma_{\mathsf{X}_j^{(1)}, \mathsf{X}_i^{(2)}}}}$, for any $i, j$ with $\Sigma_{\mathsf{X}_j^{(1)}, \mathsf{X}_i^{(2)}} \neq 0$;

- Measuring weights: $\hat{\omega}_i^{(2)} = \dfrac{\Sigma_{\mathsf{X}_i^{(1)}, \mathsf{X}_i^{(2)}}}{\hat{\omega}_i^{(1)}}$;

- Measuring noise variances: $\hat{\phi}_i^{(k)} = \Sigma_{\mathsf{X}_i^{(k)}, \mathsf{X}_i^{(k)}} - (\hat{\omega}_i^{(k)})^2$, for $k = 1, 2$, and

- Measuring noise means: $\hat{\mu}_i^{(k)} = \mathbb{E}(\mathsf{X}_i^{(k)})$, for $k = 1, 2$.

With these closed-form solutions, we have: the ratio between each latent variable's two measuring linear coefficients are identified, i.e., $\frac{\hat{\omega}^{(1)}}{\omega^{(1)}} = \frac{\hat{\omega}^{(2)}}{\omega^{(2)}} =: c \in \mathbb{R}^m$, where the division is element-wise. The latent covariance matrix is also identified to this scaling, satisfying $\Sigma_{\mathbf{L}, \mathbf{L}} = \text{diag}(c) \hat{\Sigma}_{\mathbf{L}, \mathbf{L}} \text{diag}(c)$. The means are identified to the shifting, i.e., $\omega^{(k)} * \mu_{\mathbf{L}} + \mu^{(k)} = \hat{\mu}^{(k)}$ holds for $k = 1, 2$. And last, measuring noise variances' exact values are identified, i.e., $\phi^{(k)} = \hat{\phi}^{(k)}$ holds for $k = 1, 2$.

Finally, with the identified model parameters (up to their indeterminacies) from each domain, we can compare them and identify the changing variables. Under faithfulness assumption, a latent variable $\mathsf{L}_i$ is changing, if and only if for all subsets $\mathbf{L}_C \subseteq \mathbf{L} \setminus \{\mathsf{L}_i\}$, the conditional distribution $\{p(\mathsf{L}_i | \mathbf{L}_C, \mathsf{T} = t)\}$ is changing with $\mathsf{T}$. Using second-order information, this conditional distribution can be characterized by the conditional variance $\text{var}(\mathsf{L}_i | \mathbf{L}_C)$, and regression coefficients and intercept of $\mathsf{L}_i$ on $\mathbf{L}_C$. Assumption *(A2)* ensures that these second-order information is changed.

Then, from the estimated rescaled and shifted model parameters, with assumption *(A1)*, each latent variable $\mathsf{L}_i$ must have at least one measurement with invariant linear coefficients $\omega_i^{(k)}$ and means $\mu_i^{(k)}$ across domains. For each $\mathsf{L}_i$, its unknown corresponding invariant measurement can be identified as follows: for each $\mathsf{L}_i$, pick one measurement $\mathsf{X}_i^{(k)}$ as if it was invariant, and rescale and shift $\mathsf{L}_i$ from different domains so that $\hat{\omega}_i^{(k)}$ and $\hat{\mu}_i^{(k)}$ are the same across all domains. Then, use the parameters calibrated on this set of invariant measurements to determine and to count the number of "changes". Due to the minimal change principle, which is another way to put faithfulness, the choice of $\mathsf{X}_i^{(k)}$ that can achieve the minimum number of changes must correspond to the true invariant measurements, and those recovered changes must be the true changes. This is because, when parameters are calibrated on incorrectly specified, actually changing measurements, some other truly invariant parameters will be scaled/shifted to be changing, while for the true changes, they cannot be offset to invariances. $\qquad \square$

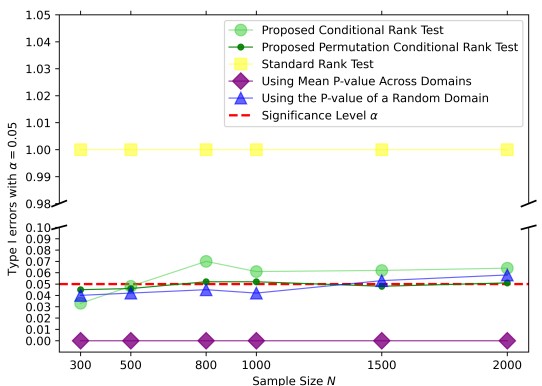
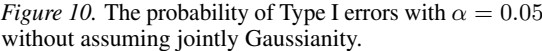

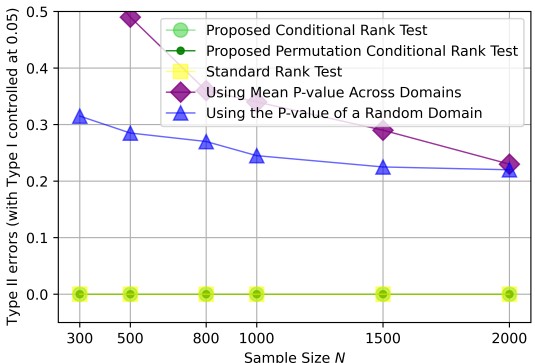

*Figure 10.* The probability of Type I errors with $\alpha = 0.05$, without assuming jointly Gaussianity.

*Figure 11.* Type II errors (effective Type I controlled at 0.05), without assuming jointly Gaussianity.

## B. Additional Experimental Results

### B.1. Experimental Results for the Proposed Conditional Rank Test

In this section we empirically validate our proposed conditional rank test from two perspectives. 1. whether the proposed test is a valid test, by checking whether it can control the Type-I error properly at a designated significance level. 2. the power of the test, by comparing the Type-II error with baselines. We consider two settings. (i) All the noise terms $g_i(\mathsf{T}, \epsilon_i)$ in Definition 1 follow Gaussian distributions and all observed variables $\mathbf{X}$ are jointly Gaussian conditioned on $\mathsf{T}$ (required by Theorem 4). (ii) All the $\epsilon_i$ follows either uniform or Laplace distributions and $h_i$ and $g_i$ are randomly initialized polynomial functions parameterized by neural networks. Consequently, the observed variables $\mathbf{X}$ generally follow non-Gaussian distributions conditioned on $\mathsf{T}$.

As for Type-I error, the results are shown in Figure 8 (where $\mathbf{X}$ are jointly Gaussian conditioned on $\mathsf{T}$) and Figure 10 (where $\mathbf{X}$ are generally non-Gaussian). Specifically, under the Gaussianity setting, with significance level 0.05, both the proposed conditional rank tests can consistently control the Type-I errors around 0.05 with varying sample sizes, which shows that the proposed test is valid. Under the non-Gaussian setting, interestingly, the test proposed in Theorem 4 can still control the Type-I well, thought the permutation test in Theorem 5 controls the Type-I better (which is as expected as the permutation test does not require Gaussianity). Plus, if we use the standard rank test (which tests the unconditional rank), the Type-I error will be very large and close to 1. This also validates the motivation of our proposed novel conditional rank tests for causal discovery in the nonstationary setting as directly plugging in a standard rank test will overly reject null hypotheses.

As for Type-II error, the results are shown in Figure 9 (where $\mathbf{X}$ are jointly Gaussian conditioned on $\mathsf{T}$) and Figure 11 (where $\mathbf{X}$ are generally non-Gaussian). As illustrated in the figures, the proposed conditional rank tests can control the Type-II error very well, even when the joint Gaussianity is violated (as in Figure 11). We also note that both standard rank test and the proposed conditional rank tests have a nearly zero Type-II error, even with a pretty small sample size ($N = 300$). Note that, thought the standard rank test also achieves a good Type-II error result, it is at the cost of having a very high Type-I error, i.e., by overly rejecting even the true null hypothesis. In contrast. the proposed conditional rank test not only properly controls the Type-I error but also controls the Type-II error very effectively.

### B.2. Results on Real-World Big Five Personality Dataset

In this section, we aim to employ the real-world Big Five personality dataset (`openpsychometrics.org`) to validate the proposed LCD-NOD method. This dataset contains 50 personality indicators / questions with 19,719 datapoints. Each data point corresponds to a person that participates the questionnaire and each indicator's value is the response ("Disagree", "Slightly Disagree", "Neutral", "Slightly Agree", "Agree") of the person to each question (e.g., "I am the life of the party"). Psychologists believe that there are five major dimensions that underlie human personality: Openness, Conscientiousness, Extraversion, Agreeableness, and Neuroticism (O-C-E-A-N), and in the dataset, each dimension is designed to be measured by 10 questions (e.g., O1 is the first question for Openness). In this section we only use 24 out of the 50 questions to make the main conclusion of the result clearer. Further. the dataset contains the country information of the participants, which is taken as the domain index $\mathsf{T}$ in our experiment. As the number of data points for some countries could be very small, we choose the top-6 common countries in the dataset to produce the structure result, which are United States, United Kingdom,

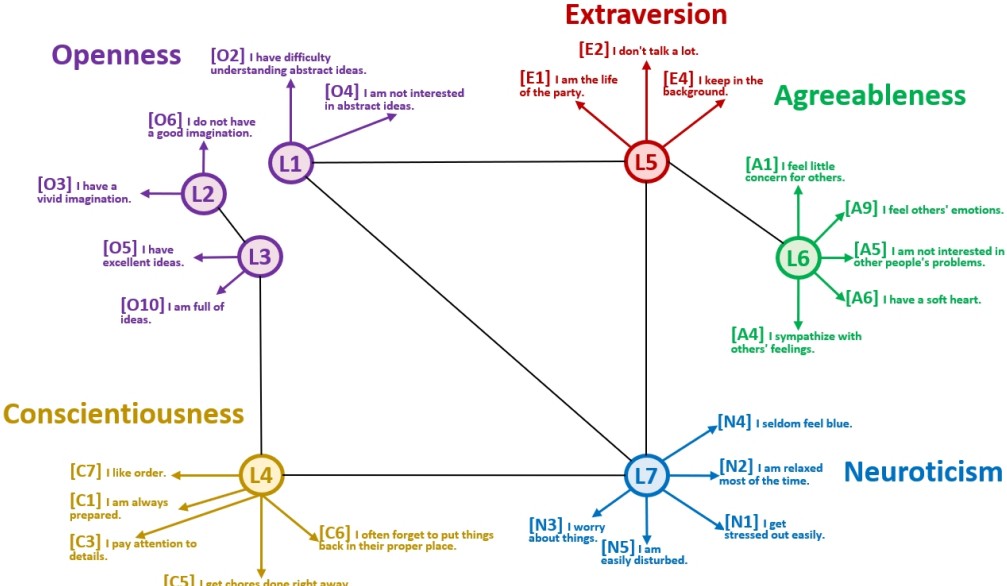

*Figure 12.* Causal structure (CPDAG) on Big Five personality data found by Phase 1 of LCD-NOD.

Canada, Australia, Philippines, and India. The structure produced by Phase 1 of LCD-NOD (based on FOFC) is shown in Figure 12 and the augmented structure with country by Phase 2 of LCD-NOD is shown in Figure 13.

As we can see, without any prior knowledge, the structure recovered by Phase 1 of our method aligns well with existing psychometric studies: each item in our result is indeed caused by the corresponding Big Five dimension, e.g., N1,...,N5 are all caused by the same latent variable L7, which is expected to correspond to Neuroticism. This result empirically validates the effectiveness of LCD-NOD from a psychological perspective.

We also found that, by using LCD-NOD with proposed conditional rank test, we can discovery meaningful structure with a reasonable significance level (1e-4); as a comparison, previous rank based methods such as hier-rank (Huang et al., 2022) and RLCD (Dong et al., 2024a) have to use extremely small significance level (smaller than 1e-10) to produce plausible results. The reason lies in that, the underlying human personality model is heterogeneous and how variables affect each other varies across different countries. The previous rank based methods fail to consider and deal with such nonstationarity and their test p-values correspond to unconditional rank that cannot correctly reflect the desired t-separations. In contrast, by leveraging the proposed conditional rank test, LCD-NOD does not suffer from this issue.

Further, the Phase 2 of LCD-NOD can be leveraged to discovery how the underlying causal model changes across different countries. The corresponding result can be found in Figure 13, where the edge from country to a variable, say, $V_i$, means that $h_{:,i}$ and $g_i$ changes with country. Take L3 in Figure 13 as an example. The existence of edge from country to L3 informs us that, although for all the countries, L4 (corresponds to conscientiousness) has a positive effect on L3 (a sub-dimension of openness that corresponds to ideas), the strength of such influence varies across countries. We further look into the edge coefficients and found that in India, the strength of the edge L4→L3 is around +0.45, which means a very strong causal influence from conscientiousness to ideativeness; on the contrary, in all other five countries this causal strength from L4 to L3 is only around 0.1. This informs us that, the underlying causal model related to variable L3 is indeed nonstationary across countries and especially different in India.

We also analyze the sensitivity of the Phase 2 result on the Big Five dataset to the choice of the countries. Specifically, we produce the result using the top-15 countries instead and found that the discovered structure is unchanged and the found changing variables are nearly identical, suggesting the conclusion is reasonably robust to the choice of country set.

**B.3. Disentangled Result for Phase 2**

In this section we take the ground truth structure of the output of Phase 1 as the input of Phase 2. This is because, though in the large sample limit, the phase 1 can produce the correct structure, given finite samples, there always exists statistical

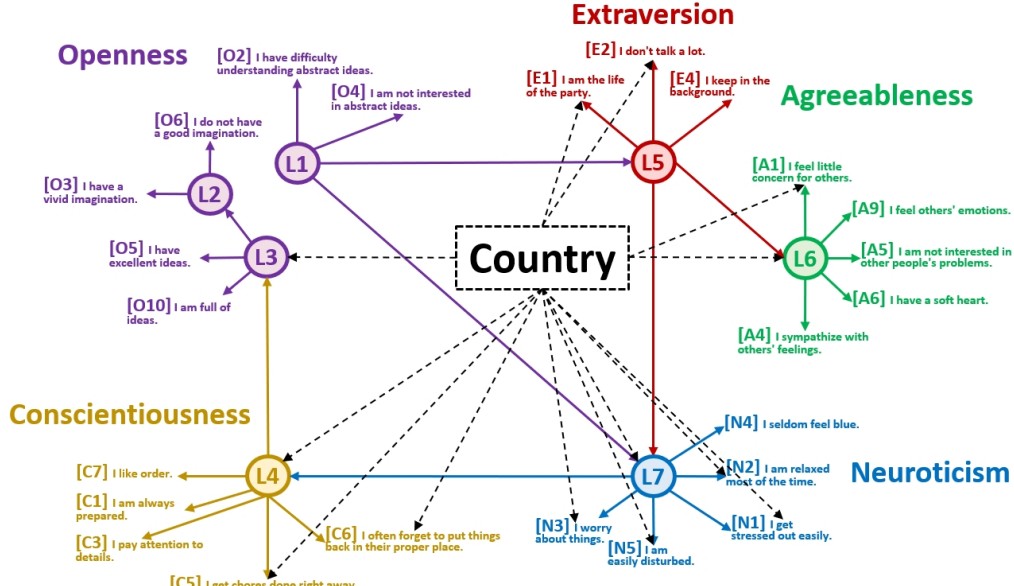

*Figure 13.* The augmented structure (structure with domain index Country) on Big Five found by Phase 2 of LCD-NOD.

errors. Thus this setting can test the performance of Phase 2 without the influence from the statistical errors in Phase 1. The result is shown in Figure 16 and Figure 17. As expected, LCD-NOD achieves better performance than that in the end-to-end setting.

We also conduct robustness analysis on to what extent the empirical performance of LCD-NOD relies on the assumption that $g$ and $h$ are independent given $\mathsf{T}$. Specifically we add a shared term $k\mathsf{C}$ to both $g$ and $h$ where $k$ controls the strength of the dependence between them given $\mathsf{T}$. We found that under mild misspecification (k=0.1) LCD-NOD remains fairly robust (Phase 1 with 10000 samples achieves 0.85 F1 and decreases only to 0.81 under this misspecification); performance degrades as the dependence becomes stronger (e.g., k=0.5), which is as expected.

### B.4. Comparison with LIT (Yang et al., 2024) and UT-IGSP (Squires et al., 2020)

Here we restrict the setting to accommodate the constraints of LIT and UT-IGSP. Specifically we allow changes only in exogenous noises and identifying edges only from $\mathsf{T}$ to $\mathsf{X}$, and compare the **end-to-end** result of Phase 2 of LCD-NOD with LIT and UT-IGSP. The result is shown in Table 2. As shown in the table, LCD-NOD still consistently outperforms them across different sample sizes though the setting is in favor of them.

A conceptual comparison with LIT's identi- fiability with LCD-NOD is as follows. LIT's Thm 2 shows LIT returns a superset of the targets; when changing latent variables exist, this superset can be substantially larger than the true one and may be uninformative. In contrast, Phase 2 of LCD-NOD aims to identify exactly which latent and observed variables are changing.

For example, consider a one-factor model with latent variables $\mathsf{L}_1 \to \mathsf{L}_2$, where $\mathsf{L}_1$ and $\mathsf{L}_2$ has observed children $\{\mathsf{X}_1^1, \mathsf{X}_1^2, \mathsf{X}_1^3\}$, $\{\mathsf{X}_2^1, \mathsf{X}_2^2, \mathsf{X}_2^3\}$ respectively. Suppose $\mathsf{L}_1$ and $\mathsf{X}_1^1$ change. In LIT's framework, as the changing latent $\mathsf{L}_1$ is the ancestor, the influence propagates to all observed variables, and thus LIT generally returns the set of all observed variables. At the same time, Phase 2 of LCD-NOD can locate the changing variables specifically as $\mathsf{L}_1$ and $\mathsf{X}_1^1$.

Yet, we do not claim Phase 2 of LCD-NOD is uniformly more general. Rather, they make different trade-offs: LIT does not explicitly model latents and thus does not rely on graphical assumptions but only guarantees a superset; LCD-NOD Phase 2 instead prioritizes precise localization; as it relies on the development of parameter identifiability theory, it currently specializes to the nonstationary one-factor setting but also leave a path to extend to broader structures.

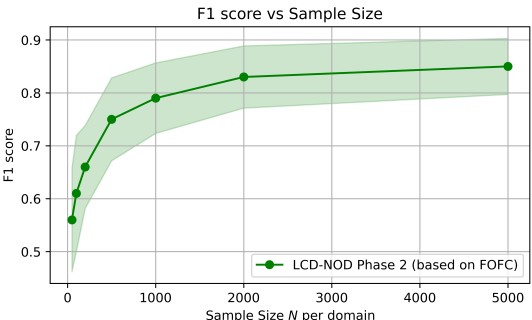

*Figure 14.* End-to-end performance of LCD-NOD Phase 2 by F1 score regarding $\mathcal{G}^{\mathrm{aug}}$ (structure involving T) of each method under OFM.

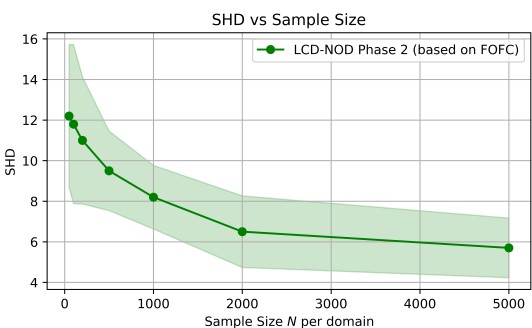

*Figure 15.* End-to-end performance of LCD-NOD Phase 2 by SHD regarding $\mathcal{G}^{\mathrm{aug}}$ (structure involving T) of each method under OFM.

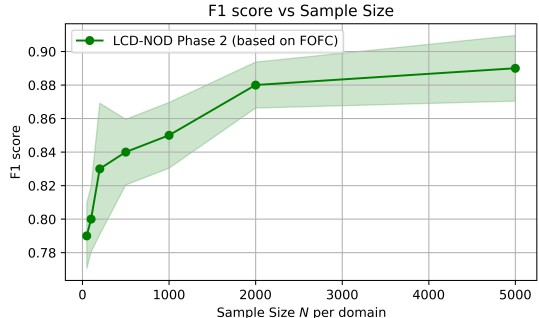

*Figure 16.* Disentangled performance of LCD-NOD Phase 2 F1 score regarding $\mathcal{G}^{\mathrm{aug}}$ (structure involving T) of each method under the OFM assumption with 95% Confidence Interval.

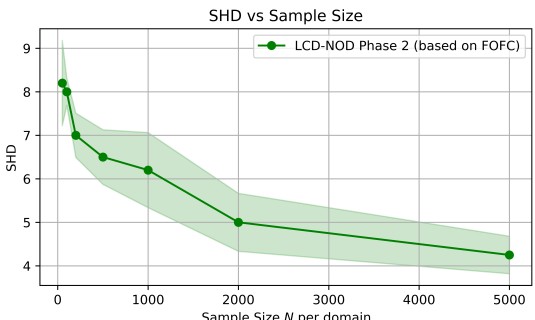

*Figure 17.* Disentangled performance of LCD-NOD Phase 2 by SHD regarding $\mathcal{G}^{\mathrm{aug}}$ (structure involving T) of each method under the OFM assumption with 95% Confidence Interval.

## C. Additional Information

### C.1. The Rationale Behind Definition 1

The rationale behind the design of Definition 1 is that: (1) we want a model class where we can establish identifiability when latent variables and nonstationarity coexist, and (2) subject to (1), the modeling of nonstationarity should be as general as possible.

A natural starting point is from linear causal models and further allow both edge coefficients and exogenous noises to depend on T for across-domain nonstationarity:

$$V_i = \sum_{V_j \in Pa(V_i)} h_{j,i}(\mathsf{T}) V_j + g_i(\mathsf{T}, \epsilon_i).$$

While developing the proof of Thm 1, we found that the same proof strategy still goes through if we further allow within-domain variability in $h$ through additional $\delta$, as

$$V_i = \sum_{V_j \in Pa(V_i)} h_{j,i}(\mathsf{T}, \delta_{j,i}) V_j + g_i(\mathsf{T}, \epsilon_i).$$

This led to Definition 1 in its present form,

The assumption that $g$ and $h$ are independent given T is not only required by our proof technique but also genuinely necessary.

Specifically, (1) the assumption is used to prove Lemma 9 (which is essential to the proof of Theorem 1). Specifically, let $(I - H(\mathsf{T})^\top)^{-1} = A$ and $g(\mathsf{T}) = B$, the first key step to prove Lemma 9 is

$$\Sigma_{\mathbf{V}|\mathsf{T}} = \mathbb{E}[ABB^\top A^\top | \mathsf{T}] = \mathbb{E}[A\mathbb{E}[BB^\top | \mathsf{T}]A^\top | \mathsf{T}],$$

*Table 2.* Comparison of the **end-to-end** performance of Phase 2 of LCD-NOD with LIT and UT-IGSP by F1 score with different sample sizes.

| Method | Sample size per domain | | | | | | |
|---|---|---|---|---|---|---|---|
| | N=50 | N=100 | N=200 | N=500 | N=1000 | N=2000 | N=10000 |
| LCD-NOD (end-to-end) | **0.40** | **0.41** | **0.48** | **0.57** | **0.63** | **0.69** | **0.76** |
| LIT | 0.27 | 0.24 | 0.31 | 0.28 | 0.31 | 0.32 | 0.32 |
| UT-IGSP | 0.30 | 0.36 | 0.42 | 0.39 | 0.40 | 0.39 | 0.41 |

which relies on independence of $g$ and $H$ given $\mathsf{T}$. This allow us to eliminate the randomness in $B$ and focus on handling $A$.

(2) Without this assumption, one can easily construct examples to show Theorem 1 fails (Section C.5). Roughly, when the assumption does not hold, there must exist one additional common source $C$ between $g$ and $H$ such that some equality constraints that hold in $\Theta(\mathcal{G})$ does not hold in $\Phi(\mathcal{G})$, and thus $\Phi(\mathcal{G}) \neq \Theta(\mathcal{G})$.

## C.2. Definition of Trek

**Definition 7** (Treks (Sullivant et al., 2010))**.** In $\mathcal{G}$, a trek from $\mathsf{X}$ to $\mathsf{Y}$ is an ordered pair of directed paths $(P_1, P_2)$ where $P_1$ has a sink $\mathsf{X}$, $P_2$ has a sink $\mathsf{Y}$, and both $P_1$ and $P_2$ have the same source $\mathsf{Z}$.

## C.3. Implementation Details of LCD-NOD

Phase 1 of LCD-NOD is a general augmentation of existing equality-constraint-based methods, and its implementation details involving the details for the proposed conditional rank test and the details for how to substitute the test in the original equality-constraint-based methods. As for the proposed conditional rank test, each time we first have a null hypo that the rank of $\Sigma_{\mathbf{A}, \mathbf{B}|\mathsf{T}}$ is smaller or equal to $k$, and follow Theorem 4 to calculate the test statistics $\Lambda(k)$ from observed data. Then we plug in the test statistics to the null distribution following Theorem 4 to calculate the p-value. Under a specific significance level $\alpha$, we reject the null hypothesis when the p-value is smaller than $\alpha$. In our synthetic data, we use $\alpha = 0.05$ for all the compared methods. As for how to substitute the original equality constraint test by the proposed conditional rank test, please kindly refer to Section 4.1.

For Phase 2, since we already have $\mathbf{L}$ to $\mathbf{X}$'s correspondence, theoretically we can directly solve for the parameters using the closed-form expressions provided in Section A.7. However, in practice, there might be model mis-specification and the input to the square root terms may not always be positive, i.e., some inequality constraints are not satisfied. Hence, we use MLE to estimate the parameters that produce the maximum likelihood for the observed sample covariances, though not necessarily the exact covariance values. Specifically, for each DAG over $\mathbf{L}$ consistent with the CPDAG obtained from Phase 1, we have one DAG over $\mathbf{L} \cup \mathbf{X}$. Using this DAG and observed data over $\mathbf{X}$, we identify the model parameters using the technique from (Dong et al., 2024b). Then, for each possible choice of invariant measurements, the corresponding parameters are calibrated and the changes are determined and counted. Finally, the configuration of DAGs and the set of invariant measurements that realize the minimum number of changes are identified as the equivalence of true DAGs under these changes, the true invariant measurements, and the corresponding changes are the true changes. The optimization for parameter identification is solved by Adam.

To assess convergence, we used the best log-likelihood found over a large reference run (multiple lr, 1000 restarts per lr, and 1000 iterations) as a practical reference value. We then compared smaller optimization settings against this reference best value. Empirically, we found that using 20 random restarts with lr=0.02 and 200 iterations was already sufficient: the best log-likelihood obtained under this setting was within about $0.1\%$ of the reference best value. Accordingly, this is the setting used in our experiments.

## C.4. Runtime Analysis of LCD-NOD

First we note that in LCD-NOD, the runtime is almost irrelevant to the sample size, as we only need to calculate the covariance and conditional covariance matrices once and save it for further use; the result of the procedure is irrelevant to sample size in terms of time complexity.

For Phase 1, the time complexity depends on two things: the complexity of the proposed conditional rank test and the complexity of the to-be-upgraded baseline method. As for the rank test, the time complexity of the standard rank test for $\Sigma_{\mathbf{X}, \mathbf{Y}}$ is $\mathcal{O}(\max(|\mathbf{X}|, |\mathbf{Y}|)^3)$ and the complexity of the proposed conditional rank test is $\mathcal{O}(|\mathrm{supp}(\mathsf{T})| \times \max(|\mathbf{X}|, |\mathbf{Y}|)^3)$,

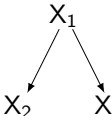

*(a)* The causal graph for a model that satisfies Definition 1.

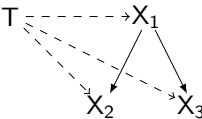

*(b)* $\mathcal{G}^{\mathrm{aug}}$ under nonstationary setting where the dashed arrows from T to variables means $g$ and $h$ are funcstion of T.

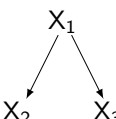

*(c)* The causal graph for a model where $g$ and $h$ are functions of both T and C.

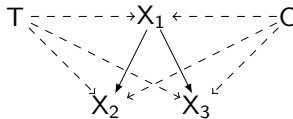

*(d)* The corresponding causal graph that explicitly includes T and C, where $X_1$, T, and C are all the confounders of $X_2, X_3$, and thus the constraint that $X_2 \perp\!\!\!\perp X_3 | T, X_1$ does not hold any more.

*Figure 18.* Illustrative example to show that, if $g$ and $h$ are not independent given T, the constraints on the conditional covariance matrix (conditioned on T) will be different.

where $|\mathrm{supp}(T)|$ refers to the number of different values that T can take. This is because we need to conditioned on each value of T to calculate the likelihood ratio statistics as in the proposed Theorem 4. We note that there is a trade-off between time complexity and test power. As likelihood ratio test is often asymptotically optimal in terms of power, we cannot further optimize the time complexity without compromising the test power. As a comparison, if we randomly choose a domain (a value of T), then we can also build a naive but valid conditional rank test that has the same complexity as that of the standard rank test; Yet, the power is not as good as the proposed likelihood ratio based conditional rank test (as shown in Figures 8 to 11 ). The time complexity for the to-be-upgraded equality-constraint-based methods, e.g., PC and RLCD, varies and highly depends on the number of variables and the sparsity of the ground truth graph. Thus, Specifically, in the worst case they have a complexity exponential in the number of variables. However, if the underlying graph is sparse, which is a common and reasonable assumption (Kalisch et al. 2007), the complexity becomes polynomial. In our empirical experiments with single Intel(R) Xeon(R) CPU E5-2470, the Phase 1 of LCD-NOD is pretty fast and it takes only around 10 seconds, 3 seconds, and 30 seconds, per graph with average 15 nodes for LCD-NOD (PC-based), LCD-NOD (FOFC-based), and LCD-NOD (RLCD-based) respectively.

For Phase 2, the time complexity depends on the number of MLE parameter estimations needed, that is, the number of DAGs over $\mathbf{L}$ consistent with the CPDAG estimated from Phase 1. This traversing process can be done in $\mathcal{O}(|\mathbf{L}|^4)$ using clique-picking and memorization (Wienobst et al. 2023). On each of the DAG, time complexity for parameter estimation is $\mathcal{O}(t|\mathbf{V}|^3)$, where $|\mathbf{V}|$ is the number of variables in the graph and $t$ is the number of iterations of gradient descent (Sivan et al. 1997). Then finally, to choose the invariant measurement configuration, since all parameters are already identified up to indeterminacies, this becomes a simple rescaling of matrices and can be done at once by broadcasting the rescaling operations under different choices all directly to one tensor operation. In practice, since MLE with latent variables is non-convex and there may be local solutions, for each DAG we run the MLE estimation procedure under different learning rates and multiple restarts and choose the one with best likelihood. Even under this, for the O-C-E-A-N real dataset with 7 latent variables and 25 measurements, the whole time for identifying changes is less than 30 seconds, under the same experimental environment as mentioned above.

### C.5. Illustrative Example to show The Relation between $\Theta(\mathcal{G})$ and $\Phi(\mathcal{G})$ Does not hold When $g$ and $h$ are not independent given T.

The example is shown in Figure 18. Specifically, (a) The causal graph of a model that satisfies Definition 1: $h_{j,i} = h_{j,i}(T, \delta_{j,i})$ and $g_i = g_i(T, \epsilon_i)$. Thus $g$ and $h$ are independent given T. (b) The causal graph that explicitly includes T, where $X_2 \perp\!\!\!\perp X_3 | T, X_1$ holds. (c) The same causal graph as (a) but the model does not satisfies Definition 1. Specifically, $h_{j,i} = h_{j,i}(T, \delta_{j,i}, C)$ and $g_i = g_i(T, \epsilon_i, C)$. Thus, $g$ and $h$ are not independent given T. (d) The effective causal graph that explicitly includes T and C, where $X_1$, T, and C are all the confounders of $X_2, X_3$, and thus the constraint $X_2 \perp\!\!\!\perp X_3 | T, X_1$ does not hold any more.

## D. Limitations

One limitation of this work is that, our proposed conditional rank test in Theorem 4 has to assume that all variables $\mathbf{X}$ are jointly gaussian given $\mathsf{T}$; otherwise it is very difficult to derive the null distribution. However, we note that this is a common limitation of existing rank test, as the standard rank test also has to assume jointly gaussian. Plus, our empirical result in Section B.1 empirically shows that, even when the data is not jointly gaussian, the proposed method can still control the Type-I properly and control the Type-II error effectively. In addition, to further address this limit, we also proposed a permutation-based conditional rank test, as in Theorem 5, which properly controls the Type-I errors without requiring joint Gaussianity.

