# OpenReview forum: "Identifying Partially Observed Causal Models from Heterogeneous/Nonstationary Data"
_ICML.cc/2026/Conference — ICML 2026 regular_

### Official Review · Reviewer_BMPm · 2026-02-27

**Soundness:** 3
**Presentation:** 3
**Significance:** 3
**Originality:** 3
**Overall Recommendation:** 3
**Confidence:** 3

**Summary:**

This paper proposes LCD-NOD, a method for causal structure identification in partially observed linear causal models with nonstationary or heterogeneous data. The main theoretical contribution is Theorem 1, which establishes that distributional constraints in nonstationary POLCMs are equivalent to those in stationary POLCMs. Based on this equivalence, the authors develop novel rank tests for conditional covariance and propose a two-phase algorithm where Phase 1 identifies the causal structure G, and Phase 2 identifies variables whose causal mechanisms vary across domains.

**Compliance With Llm Reviewing Policy:**

Affirmed.

**Key Questions For Authors:**

1.What is the rationality of nonstationary POLCMs in definition 1? Please give more explanations.

2.In Section 2.1, the authors refer to Theorem 1 twice, but Theorem 1 itself is presented only later.

3.Please correct the citation format in typo and ensure the first letter of journal names is capitalized.

4.How many rank tests does LCD-NOD actually perform on your experimental problems, and what is the resulting family-wise error rate?

5.Given the non-convex nature of Phase 2's MLE optimization, can you provide convergence diagnostics or guarantees demonstrating that reported solutions are close to global optima?

6.How sensitive are your qualitative conclusions about Big Five structure changes to the selection of only top-6 countries? Would including all countries substantially alter the detected mechanisms?

**Limitations:**

yes.

**Strengths And Weaknesses:**

Strengths
1. The paper addresses an important gap by simultaneously handling latent confounders and nonstationarity, two challenges that often appear together in real-world applications.
2. The core theoretical result, Theorem 1, is conceptually elegant and provides a principled justification for upgrading existing constraint-based causal discovery methods to the nonstationary setting.
3. The framework demonstrates good generality since Phase 1 can enhance multiple baseline methods including PC, FOFC, and RLCD, showing strong modularity.
4. The experimental evaluation is comprehensive, including synthetic data validation with ablation studies and a real-world application on Big Five personality data.
5. Theorem 5 provides a valuable permutation-based rank test that does not require the Gaussianity assumption, thereby extending the method's applicability.

Weakness

1.LCD-NOD performs numerous rank tests across different variable pairs and rank thresholds k, yet the paper does not address the multiple testing problem.

2.Theorem 1 is the cornerstone of the paper, yet its proof reveals several gaps. Definition 1 implicitly assumes a specific independence structure among noise terms (δ, ε relationships) without explicitly stating these assumptions, which are crucial for the proof. The critical step in Lemma 9 (transition from Equation 11 to 12) relies on independence exchangeability for expectations, but the formal justification for applying this to random coefficients H(T,δ) rather than deterministic coefficients is absent. The use of Trek decomposition for random coefficients needs substantially stronger justification beyond what the standard deterministic case provides.

3.Theorem 4's convergence analysis assumes that sample sizes N_t → ∞ for all t ∈ supp(T), yet practical datasets frequently exhibit imbalanced domains with vastly different sample sizes. The paper provides no discussion of how such imbalance affects the asymptotic distribution or at what sample sizes the approximation becomes reliable.

4.Theorem 5 attempts to provide a distribution-free rank test through permutation, but the proof confuses graphical and statistical independence concepts. The argument that rank faithfulness implies asymptotic exchangeability requires more careful exposition.

5.Phase 2 relies on maximum likelihood estimation for latent variable models, which is inherently highly non-convex. The paper mentions using "different learning rates and multiple restarts" but provides no practical guidance. How many restarts are needed, what learning rate schedule should be used, and how should convergence be assessed?

6.The exact DAG structures, parameter ranges, and configuration of domains and sample sizes are either unspecified or incompletely described. The "PC/FOFC/RLCD with conditioned data" baseline is vaguely defined, making it unclear exactly what algorithm is being compared.

7.Important robustness analyses are completely missing. There is no investigation of what happens when key assumptions are violated, such as when g and h are not independent given T. No scalability study examines how the method performs on high-dimensional problems with over 100 variables. The Big Five personality dataset experiment lacks ground truth for validation; the claim that results "align with psychology" is entirely qualitative and subjective. Furthermore, the authors restrict analysis to the top-6 countries due to sample size constraints, introducing selection bias that is never discussed.

---

> ### Author Rebuttal · Authors · 2026-03-31
>
> Thank you for the positive assessment of the motivation, the conceptual role of Thm 1, the generality of Phase 1, and the value of the permutation test in Thm 5. We agree that the presentation can be improved, and we have revised accordingly.
>
> **1. Rationale of Def 1**
>
> Please refer to **1.** in our response to reviewer b7XB.
>
> **2. Critical steps in Lemma 9**
>
> Thank you for pointing out that this part was too compressed. We have revised the proof to make the following explicit.
>
> (1) The step from Eq 11 to 12 **does not treat random coefficients as deterministic**. It only uses all h are mutually independent given $\mathsf{T}$.  The detailed derivation:
> $$\mathbb{E} [\sum \Omega_{g\text{top}(P_1,P_2)}H^{P_1}H^{P_2}|\mathsf{T}] = \sum\Omega_{g\text{top}(P_1,P_2)}\mathbb{E}[H^{P_1}H^{P_2}|\mathsf{T}].$$
> As the structure is acyclic, there is no repeated edges in $P_1P_2$. Thus $H^{P_1}H^{P_2}$ is a multiplication of independent terms given $\mathsf{T}$ and thus
> $$=\sum\Omega_{g\text{top}(P_1,P_2)}\mathbb{E}[H|\mathsf{T}]^{P_1}\mathbb{E}[H|\mathsf{T}]^{P_2}.$$
>
> (2) In the integration form of the expectation, the trek expansion is applied pointwise to **each realization of $H$**.
> Thus the trek decomposition itself is purely algebraic, exactly as in the deterministic case.
>
> **3. FWER, number of tests, multiple testing**
>
> We agree this should be clarified. As in other constraint-based methods, the tests in LCD-NOD are generally dependent, so the overall FWER cannot be computed as if they were independent. Without strong assumptions, one can only give a upper bound such as Bonferroni $FWER\le m\alpha$.
>
> This bound is typically very conservative (Wadhwa et al.  2021). For this reason, the standard and more informative evaluation is the final graph recovery performance over repeated simulations, which also capture the cumulative effect of all tests. We will clarify this and report the empirical number of tests in our experiments (roughly 2000 for LCD-NOD+PC and 5000 for LCD-NOD+RLCD per run).
>
> Importantly, Phase 1 of LCD-NOD does not introduce an additional layer of multiple testing beyond the base method; it only replaces the tests.
>
> **4. Thm 4 under imbalanced domains**
>
> Yes the asymptotic approximation requires sufficient sample size in each domain. We have revised to state this more clearly. In practice, Thm 4 is most reliable when no domain is too small; empirically, we found that having at least about 50 samples per domain is helpful for stable Type-I/-II behavior.
>
> **5. Thm 5 proof logic**
>
> We agree the proof sketch should be sharpened. Our intended logic is:
>
> (i) under the null, after SVD rotation, the trailing canonical coordinates have asymptotically zero cross-covariance within each domain;
>
> (ii) by Thm 1, each domain is equivalent, at the level of second-order constraints, to a stationary linear SEM with the same graph;
>
> (iii) rank faithfulness is used only to rule out degenerate cancellations, so these zero cross-covariances reflect genuine absence of shared exogenous sources in the two rotated trailing coordinates;
>
> (iv) therefore the two rotated trailing coordinates are functions of disjoint sets of independent noises and are thus asymptotically independent within each domain.
>
> We have revised the appendix to make this argument explicit.
>
> **6. Phase 2 optimization details and convergence diagnostics**
>
> Please refer to **4.** in our response to reviewer EH3k.
>
> **7. Experimental details**
>
> We agree the details can be made clearer. We have revised the paper by: (i) listing the graph-generation settings and parameter ranges in one place, and (ii) defining the “conditioned-data” baselines more explicitly.
>
> **8. Robustness, scalability, and Big Five**
>
> We agree these analyses are important. We have added experiments where
> a shared term $k\mathsf{C}$ is added to both g and h and k controls the strength of the dependence between them given $\mathsf{T}$.  We found that under mild misspecification (k=0.1) LCD-NOD remains fairly robust (Phase 1 with 10000 samples achieves 0.85 F1 and decreases only to 0.81 under misspecification), though performance degrades as the dependence becomes stronger (e.g., k=0.5), which is as expected. We have also added experiments on 100-node graphs.
>
> For Big Five, we agree that the result is qualitative rather than ground-truth validation. We have soften the wording accordingly. We also ran a sensitivity analysis using the top-15 countries instead and found that the discovered structure is unchanged and the found changing variables are nearly identical, suggesting the conclusion is reasonably robust to this choice.
>
> **9. Presentation issues**
>
> Thank you for the valuable comments about that Thm 1 is referred to before it is stated and that there are capitalization
> inconsistencies.  We have corrected the issues and improved the exposition throughout.
>
> ---
>
> If any part remains unclear, please feel free to let us know. Thank you again for your valuable feedback!

---

### Official Review · Reviewer_fqW5 · 2026-03-12

**Soundness:** 2
**Presentation:** 3
**Significance:** 3
**Originality:** 3
**Overall Recommendation:** 4
**Confidence:** 3

**Summary:**

This work studies the identifiability of causal graph structures in non-stationary datasets with latent variables. The authors first provide an equivalence relation between the covariance matrices of a non-stationary linear causal structural model. To combine data from multiple domains, the paper extends the rank test from single-domain to the multi-domain setting. Based on this result, the authors propose a two-phase framework that first identifies the causal graph structure and then detects the domain-variant variables. The theoretical results are validated through experiments on synthetic datasets for both phases, as well as the hypothesis test, and Phase 1 is further evaluated on a real dataset.

**Compliance With Llm Reviewing Policy:**

Affirmed.

**Final Justification:**

I decide to keep my positive score.

**Key Questions For Authors:**

1.	If I understand correctly, the motivation to provide theorem 4&5 is to utilize the samples from all domains to reduce the statistical error. Then why in Phase 1 introduced in Sec 4, only one domain data point is used?

2.	What does the union of two random vectors mean in Theorem 4? For me, it does not look like a standard way.

3.	How many latent variables are generated in the synthetic experiments? What is the relative proportion? Does the number affect the results?

**Limitations:**

yes

**Strengths And Weaknesses:**

## Strength

1.Novelty and Significance: Studying extensions of the rank test to efficiently handle non-stationary data is both novel and important. To the best of my knowledge, identifying causal graphs in the presence of latent variables remains a challenging problem. This work takes a meaningful step toward addressing causal structure recovery in such settings.

2.Presentation: The overall structure of the paper is clear and easy to follow. The authors provide intuition through a concrete example, and the explanations following each theorem are helpful for guiding the reading through the theoretical results.

## Weakness

1.	Some notation usage are not standard. For example, in Theorem 4, it looks unusual for me to use |A| for the cardinality of random vector (which I guess it means based on the context. Also, the use of the union operator in this context is unclear (as also mentioned in the questions below). Clarifying these notations would improve readability.

2.	Limited significance of Phase 2.

From theoretical view, Assumption 7 is somewhat restrictive because it requires measurement invariance across domains and has no clear way to verify it with latent variables involved.

Although the theory only requires one invariant measurement per latent variable, in practice this may be difficult to identify due to numerical estimation errors, especially when the sample size within each domain is small. As a result, it may be challenging in empirical settings to reliably detect such an invariant “anchor” coefficient.


3.	Soundness of empirical validation:

The experiment's evaluation overall is sufficient. The author evaluates each phase separately, as well as end-to-end performance of the full framework. However, experiments including both Gaussian and non-Gaussian noise settings would provide stronger empirical support and are currently missing from the main paper. In addition, since the presence of latent variables is one of the main challenges addressed by this work, it would be informative to evaluate how the non-stationarity detection performs separately on latent variables and observed variables.

---

> ### Author Rebuttal · Authors · 2026-03-31
>
> Thank you for the encouraging comments on the novelty, significance, and organization of the paper. We also appreciate the constructive suggestions that help us improve the clarity and empirical coverage.
>
> **1. Notation in Thms 4 and 5**
>
> Here $\mathbf{A}$ denotes a **set of variables**, $|\mathbf{A}|$ refer to the cardinality, and $\cup$ denotes the union of two sets. Thank you for pointing this out and we have replaced it by $\dim()$ and explicitly defined the union in the revision.
>
> **2. Phase 1 use only one domain?**
>
> The intended Phase 1 uses data from all domains. We believe the confusion comes from a typo in Sec 4.1 and we have corrected it.
>
> **3.  Phase 2 scope and path to extend**
>
> Phase 1 is the main general contribution: it gives a broad augmentation principle for existing equality-constraint-based methods under nonstationarity with identifiability guarantee. Phase 2 is narrower because it requires not only structure identifiability, but also **parameter identifiability** in latent-variable models.
>
> For one-factor models, change localization is possible as the parameters are identifiable up to only trivial (scaling) indeterminacies (Bollen, "Structural Equations ...", 1989). For more general multi-factor models, however, it has been proved that parameters are typically identifiable only up to orthogonal transformations (Dong et al., "On the Parameter ...", 2024); under such indeterminacies, changes across domains may be fundamentally undetectable without further assumptions. Thus, the narrower scope of Phase 2 is due to a genuine identifiability theory boundary rather than merely an implementation choice.
>
> That said, extending Phase 2 beyond one-factor models is still possible; the main strategy could be similar but it requires additional assumptions or new techniques to handle the orthogonal indeterminacies.  We agree that it is an important future direction. Thank you for the insightful question and we have added a related discussion to better position the contribution.
>
> **4. Need an invariant measurement anchor**
>
> The invariant-anchor assumption is needed to resolve that, the effective measurement coefficients are only identifiable up to scaling; without at least one invariant anchor per latent variable, one cannot determine whether the latent variable itself changed or whether the change is due to an arbitrary rescaling (line 336).
>
> We also agree that selecting such an anchor can be statistically challenging especially with small per-domain sample sizes. We have revised the paper to emphasize both points more clearly: (i) the assumption is to resolve a fundamental scaling indeterminacy, and (ii) improving the robustness, e.g., by novel tests for anchor selection is an important future direction.
>
> **5. Noise type.**
>
> Thank you for the comment. Though $\epsilon_i$ is Gaussian in our main synthetic setup, the effective noises $g_i(\epsilon_i,\mathsf{T})$ are generated through random polynomial functions (line 350), and are thus generally **non-Gaussian**. We consider the general non-Gaussian setting here as the theory of Phase 1 itself does not require Gaussianity,
>
> Moreover, for the statistical tests, Thm 4 assumes Gaussianity while Thm 5 does not; accordingly, in the test evaluation (Figs. 8-11), we already include both Gaussian and non-Gaussian cases. We have revised to strengthen the presentation so the empirical support for both regimes is easier to see.
>
> **6. Separate detection performance**
>
> Thank you for the suggestion. We agree that it is useful to report the detection separately on latent and observed variables. The current paper already notes qualitatively that changes in observed variables are easier to detect with an intuitive explanation (line 415), but we did not report the quantitative breakdown. Following your suggestion, we have added this result in the revision; e.g., in Phase 2 evaluation with 5000 samples, the F1 score of detecting changing observed variables is 0.92, while that of latent variables is 0.80.
>
> **7. Latent-observed proportion**
>
> We agree that the numbers should be stated more explicitly.
> Our experiments consider three regimes: (i) no-latent-variable (about 20 observed variables), (ii) one-factor-model (about 20 observed and 5 latent variables), and (iii) more general latent structure (about 20 observed and 5 latent variables). We have made these counts explicit in the revision.
>
> Motivated by your question, we examined the effect of changing the proportion to 30 observed and 5 latent variables. We found that Phase 1 remains broadly similar, while Phase 2 improves, especially for detecting changing latent variables. This matches the intuition that Phase 2 benefits from having more measurements per latent variable, since the relevant measurement coefficients can then be estimated more accurately. We have added the corresponding results in the revision.
>
> ---
>
> If any part remains unclear please feel free to let us know. Thank you again for your valuable feedback!

---

> > ### Author Rebuttal · Reviewer_fqW5 · 2026-04-02
> >
> > Thanks for the rebuttal. My main concerns have been addressed and I would like to keep my positive score.

---

> > > ### Author Response · Authors · 2026-04-04
> > >
> > > Thank you for your insightful and constructive comments, which helped us further improve the paper. We also appreciate your positive follow-up.

---

### Official Review · Reviewer_EH3k · 2026-03-13

**Soundness:** 3
**Presentation:** 3
**Significance:** 3
**Originality:** 3
**Overall Recommendation:** 5
**Confidence:** 2

**Summary:**

This paper considers partially observable linear causal models and demonstrates that a novel class of *nonstationary* partially observable linear (*) causal models are 2nd moment wise equivalent (under time/domain label conditioning). They leverage this equivalence to investigate rank deficiency conditions in the nonstationary case and build relevant tests under slightly stronger assumptions.

**Compliance With Llm Reviewing Policy:**

Affirmed.

**Final Justification:**

I have no outstanding issues, and I retain my original recommendation to accept.

**Key Questions For Authors:**

- By setting |L| = 0, one can construct a class of fully observed LCMs that are equivalent to a nonstationary version under time conditioning. Are there any specific benefits to investigating this case, or are the results already subsumed by existing papers?
- When structured, non-stationarities can be thought as interventions, which under mild conditions can lead to a strengthened recovery class. In the model the paper proposes, this is done by contrasting the statistics for different values of T, which might be a promising future direction, if a similar argument of 2nd moment set equality holds for example.

**Limitations:**

Limitations are discussed both in-text and in a separate section.

**Strengths And Weaknesses:**

## Soundness

The methodology seems sound, however there are many parts in the proofs that I am not too familiar with.


## Presentation

The writing is good and the paper reads easily. Literature review is rather extensive and comparisons with works I am familiar with are fair.


## Significance

Nonstationarity/nonhomogeneity and partial observability/latent confounders are important problems in causal discovery, so an approach that can attack both is interesting.


## Originality

The idea of focusing on the second moment information and showing at least some models of non-homogeneity doesn't change the structural information classes is an original idea. I believe the structural conditions referred in the paper are not novel but the efficient test statistics for conditional covariance that using them in the proposed context, nonstationary, time-conditional way, requires are.

On the other hand, a lower level equivalence -- in the most extreme case, distribution level -- would be more informative in general. As noted, most causal discovery is built upon conditional independence testing which in general requires more than just 2nd moments.

---

> ### Author Rebuttal · Authors · 2026-03-31
>
> Thank you for your encouraging review comments on the paper’s novelty, significance, clarity, and technical soundness. We also appreciate your insightful questions below that help improve the clarity of our paper.
>
> **1. Any benefit to investigate $|L|=0$? Is it subsumed by existing work?**
>
> Thank you for this question. When $|L|=0$, related nonstationary causal discovery results already exist, most notably CD-NOD. Our goal here is not to claim novelty for the fully observed case in isolation, but to show our framework reduces to that setting while substantially extending it to the partially observed case.
>
> More specifically, the main contribution of LCD-NOD is to handle latent-variable structure identification under nonstationarity, which is precisely the regime not covered by previous methods such as CD-NOD. In that sense, the
> $|L|=0$ case is useful for two reasons:
> (i) it serves as a sanity check showing that our framework is compatible with fully observed setting, and (ii) it highlights our result is a strict extension in scope within the linear, equality-constraint-based setting, from fully to partially observed models.
>
> So, while the fully observed case itself is not the main novelty, it is still valuable because it clarifies how our theory connects to prior work and how the proposed framework unifies the fully and partially observed settings.
>
> **2. Nonstationarities viewed as interventions and yield stronger results?**
>
> We completely agree. The source of nonstationarities can be interpreted as interventions, and this is a promising direction.
>
> Our current model is deliberately broad: $\mathsf{T}$
> may affect both edge coefficients and exogenous terms, so it can capture many intervention-like forms of heterogeneity, e.g., hard, soft, single or multi-node interventions. The benefit of this generality is that Thm 1 applies to a wide class of nonstationary mechanisms.
>
> At the same time, we agree that **additional structure on the source of nonstationarity** could lead to stronger identification results. E.g., if one assumes single-node interventions, then pair-wise domain contrast differs in at most two nodes, which can improve structure recovery or change localization result. This is closely related to our Phase 2 and is a very interesting direction for future work.
> Thank you for the comment and we have revised the paper to make this perspective clearer and to highlight intervention-structured nonstationarity as an important future direction.
>
> **3. Distribution level equivalence possible?**
>
> We agree that a distribution-level equivalence would be stronger and more informative. However, our current scope is intentionally limited to second-moment equivalence, because this is the level at which we can obtain a clean and rigorous characterization under the proposed model.
>
> The key point is that Thm 1 concerns the equality between the homogeneous covariance set and the nonstationary conditional covariance set. This is exactly the object needed to upgrade equality-constraint-based methods such as partial-correlation, tetrad, and rank-based approaches.
>
> Notice that we allow within-domain nonstationarity through random coefficients $h$ under Def 1.
> The quadratic forms for seconder order information can nicely “average out” these undesired randomness in $h$,
> On the contrary, for higher-order moments, it typically involves additional terms induced by the nonstationarity of $h$, and these terms generally cannot be reduced to achieve the same type of equivalence as in Thm 1. Therefore, under the current model assumptions, a higher-order sense of equivalence is not established and may fail in general.
>
> We agree this is an interesting open direction, but it would likely require stronger assumptions on the form of nonstationarity. Thank you again for the question and we have revised the paper to clarify these points more explicitly.
>
> **4. Phase 2 optimization and convergence diagnostics**
>
> In our experiments, Phase 2 is optimized using Adam. To assess convergence, we used the best log-likelihood found over a large reference run (multiple lr, 1000 restarts per lr, and 1000 iterations) as a practical reference value. We then compared smaller optimization settings against this reference best value.
>
> Empirically, we found that using 20 random restarts with lr=$0.02$ and 200 iterations was already sufficient: the best log-likelihood obtained under this setting was within about $0.1\%$ of the reference best value. Accordingly, this is the setting used in the reported experiments.
>
> We have revised the paper to include (i) the detailed optimization settings, (ii) the convergence analysis just described, and (iii) a clarification that the theoretical result for Phase 2 is an identifiability result conditional on successful parameter recovery, rather than a global-optimization guarantee.
>
> ---
>
> If any part remains unclear or you have further questions, please feel free to let us know. Thank you again for your valuable feedback!

---

> > ### Author Rebuttal · Reviewer_EH3k · 2026-04-01
> >
> > No issues remain.

---

> > > ### Author Response · Authors · 2026-04-04
> > >
> > > Thank you for your thoughtful and constructive comments, which helped strengthen the paper. We also sincerely appreciate your encouraging follow-up.

---

### Official Review · Reviewer_b7XB · 2026-03-15

**Soundness:** 3
**Presentation:** 2
**Significance:** 2
**Originality:** 2
**Overall Recommendation:** 4
**Confidence:** 3

**Summary:**

The authors considered causal discovery with latent variables under heterogeneous/nonstationary data. The study addresses whether equality and rank constraints used in homogeneous partially observed linear causal models can be transferred to the nonstationary setting by conditioning on the domain index $T$. The paper formalizes Nonstationary POLCMs, proves an equivalence between the homogeneous covariance set $\Theta(G)$ and the nonstationary conditional covariance set $\Phi(G)$ (Theorem 1), extends the link between rank constraints and $t$-separation to conditional covariances (Theorem 3), introduces two conditional rank tests for heterogeneous data (Theorems 4 and 5), and proposes LCD-NOD, a two-phase procedure that first upgrades existing equality-constraint methods to the nonstationary setting and then identifies which variables are directly affected by nonstationarity in a one-factor model setting.

**Compliance With Llm Reviewing Policy:**

Affirmed.

**Final Justification:**

Overall, the rebuttal improves my understanding of the paper and addresses my main concerns sufficiently. I therefore keep my overall positive assessment unchanged.

**Key Questions For Authors:**

- The appendix provides a counterexample showing that Theorem 1 can fail if $g$ and $h$ are not independent given $T$, which is helpful. Could the authors make this point much more explicit in the main text and explain whether this assumption is mainly technical or genuinely necessary for the equivalence $\Theta(G)=\Phi(G)$? This would clarify the scope of the main theorem.
- Could the authors clarify more explicitly what is new in Theorem 3 versus what follows by combining Theorem 1 with known stationary rank/(t)-separation results? As written, it is difficult to tell whether the theorem is a new characterization or mainly a transfer argument.
- The Phase 2 result appears substantially narrower than the main framing of the paper. It is limited to a one-factor model with pure measurements and additional assumptions such as invariant measurement coefficients. Could the authors better describe the boundary between the generality of Phase 1 and the specialized scope of Phase 2?
- In Table 2, the paper compares empirically with LIT and UT-IGSP under a restricted setting, but I would appreciate a sharper conceptual comparison with LIT’s identifiability results. In particular, assuming only exogenous noise distributions change across environments, how does the present change-localization result compare with what can and cannot be identified in the presence of latent confounding? This could clarify the novelty of the Phase 2 contribution.
- Is there a path to extend the second phase beyond one-factor models with pure measurements? At present, the assumptions seem quite restrictive, and it would be useful to know whether this is a fundamental barrier or mainly a limitation of the current proof technique. A discussion of multi-factor measurement models or settings without pure measurements would strengthen the paper.

**Limitations:**

The paper should discuss the restrictive assumptions behind Phase 2, including pure measurements, one-factor structure, invariant measurement coefficients, and the gap between the theorem statement and the MLE-based practical implementation.

**Strengths And Weaknesses:**

The paper addresses an interesting problem of extending latent-variable causal discovery from homogeneous data to heterogeneous/nonstationary settings. The main idea of replacing unconditional covariance/rank constraints with conditional ones given $T$. In particular, Theorem 1 is the conceptual core of the paper, since it provides the basis for adapting existing equality-constraint methods such as PC, FOFC, and RLCD to the nonstationary setting. The paper also goes beyond a purely structural result by proposing statistical tests and an algorithmic pipeline.

That said, I have several concerns. First, while the results are potentially interesting, the presentation is dense and at times hard to follow.
On soundness, Theorem 1 appears to be the strongest part of the paper, and the appendix includes a helpful counterexample showing that the equivalence can fail without the conditional independence assumption between g and h given T. However, because this theorem is central to the overall framework, I would have liked a clearer explanation in the main text of why the equivalence holds and exactly where that assumption is used.

My main concern is with the scope of Phase 2. The change-localization result only applies in a much narrower setting with a nonstationary one-factor model with pure measurements and additional assumptions such as invariant measurement coefficients. This makes the second phase more specialized than the overall framing of the paper suggests.

The experiments are supportive but not fully convincing. The synthetic results show improvements over naive baselines and better type-I error control, but the real-data validation is mostly qualitative.

---

> ### Author Rebuttal · Authors · 2026-03-31
>
> Thank you for the insightful comments and valuable feedback.
> We are encouraged by your positive comments about the significance and relevance and grateful to the suggestions about clarity. Please see our point-by-point responses below.
>
> **1. Rationale of Def 1**
>
> The rationale behind: (1) we want a model class where we can establish identifiability when latent variables and nonstationarity coexist, and (2) subject to (1), the modeling of nonstationarity should be as general as possible.
>
> A natural starting point is from linear causal models
> and further allow both edge coefficients and exogenous noises to depend on $\mathsf{T}$ for **across-domain nonstationarity**:
> $$V_i=\sum_{V_j\in{Pa}(V_i)}h_{j,i}(\mathsf{T})V_j+g_i(\mathsf{T},\epsilon_i).$$
> While developing the proof of Thm 1, we found that the same proof strategy still goes through if we further allow **within-domain variability** in $h$ through additional $\delta$, as
> $$V_i=\sum_{V_j\in{Pa}(V_i)}h_{j,i}(\mathsf{T},\delta_{j,i})V_j+g_i(\mathsf{T},\epsilon_i).$$
> This led to Def 1 in its present form.
>
> **2. The intuition behind Thm 1**
>
> The key step in the proof of Thm 1 is to show Lemma 9:
> $$\Sigma_{\mathbf{V}|\mathsf{T}}=\mathbb{E}[(I-H(\mathsf{T})^\top)^{-1}g(\mathsf{T})g(\mathsf{T})^\top (I-H(\mathsf{T})^\top)^{-\top} |\mathsf{T}]$$
> $$=(I-\mathbb{E}[H(\mathsf{T})^\top |\mathsf{T}])^{-1} \mathbb{E}[g(\mathsf{T})g(\mathsf{T})^\top |\mathsf{T}] (I-\mathbb{E}[H(\mathsf{T})^\top |\mathsf{T}])^{-\top}.$$
>
> Specifically, though $H$ and $g$ are random matrix and vector, in the calculation of $\Sigma_{\mathbf{V}|\mathsf{T}}$, only the conditional mean of $H$ and conditional covariance of $g$ that matter. This leads to a form of $\Sigma_{\mathbf{V}|\mathsf{T}}$ very similar to $\Sigma_{\mathbf{V}}$ of a homogeneous case, and thus equivalence follows.
>
> **3. Where independence of g and h given T used? Mainly technical or genuinely necessary?**
>
> This assumption is not only required by our proof technique but also genuinely necessary.
>
> (1) The assumption is used to prove Lemma 9 (which is essential to Thm 1's proof). Specifically,
> let $(I-H(\mathsf{T})^\top)^{-1}=A$ and $g(\mathsf{T})=B$,
> the first key step to prove Lemma 9 is
> $$\Sigma_{\mathbf{V}|\mathsf{T}}=\mathbb{E}[ ABB^{\top}A^{\top}|\mathsf{T}]=\mathbb{E}[ A\mathbb{E}[BB^{\top}|\mathsf{T}]A^{\top}|\mathsf{T}],$$
> which relies on independence of $g$ and $H$ given $\mathsf{T}$. This allow us to eliminate $B$'s randomness and focus on handling $A$.
>
> (2) Without this assumption, one can easily construct examples to show Thm 1 fails (Appx C.4).
> Roughly, when the assumption does not hold, there must exist one additional common source $C$ between $g$ and $H$
> such that some equality constraints that hold in $\Theta(\mathcal{G})$ does not hold in $\Phi(\mathcal{G})$, and thus $\Phi(\mathcal{G})\neq \Theta(\mathcal{G})$.
>
> **4. Is Thm 3 a new characterization or a transfer argument?**
>
> Thm 3 is a nonstationary transfer result from Thm 1, In light of your question, we have revised it as a Corollary to avoid overstating.
>
> **5. Generality of Phase1 and specialized scope of Phase2? Path to extend?**
>
> Please refer to **3.** in our response to reviewer fqW5.
>
> **6. A sharper conceptual comparison with LIT’s identifiability (assuming latent variables and only exogenous noise change)?**
>
> Thanks for the insightful question. In this setting, LIT's Thm 2 shows LIT returns a **superset of the targets**; when changing latent variables exist, this superset can be substantially larger than the true one and may be uninformative.
> In contrast, Phase 2 of LCD-NOD aims to identify **exactly which latent and observed variables** are changing.
>
> For example, consider a one-factor model with latent variables$L_1\to L_2$, where$L_1$ and$L_2$ has observed children $\\{X_1^1,X_1^2,X_1^3\\},\\{X_2^1,X_2^2,X_2^3\\}$ respectively. Suppose $L_1$ and $X_1^1$ change. In LIT’s framework, as the changing latent $L_1$ is the ancestor, the influence propagates to all observed variables, and thus LIT  generally returns the set of all observed variables. At the same time, Phase 2 of LCD-NOD can locate the changing variables specifically as $L_1$ and $X_1^1$.
>
> Plus, this precise localization does not require that only exogenous noise changes. Yet, we do not claim Phase 2 is uniformly more general. Rather, they make different trade-offs: LIT does not explicitly model latents and thus does not rely on graphical assumptions but only guarantees a superset; LCD-NOD Phase 2 instead prioritizes precise localization; as it relies on the development of parameter identifiability theory (see **5.**), it currently specializes to the nonstationary one-factor setting but also leave a path to extend to broader structures.
>
> **7. The gap between theorem and MLE implementation?**
>
> Please refer to **4.** in our response to reviewer EH3k.
>
> ---
>
> Thank you again for your valuable feedback. If any part remains unclear or you have further questions,
> please feel free to let us know.

---

> > ### Author Rebuttal · Reviewer_b7XB · 2026-04-01
> >
> > Thanks for the detailed rebuttal. Overall, the rebuttal improves my understanding of the paper and addresses my main concerns sufficiently. I therefore keep my overall positive assessment unchanged.

---

> > > ### Author Response · Authors · 2026-04-04
> > >
> > > Thank you for your prompt and constructive comments, which helped improve the quality of our paper. We also greatly appreciate your encouraging follow-up and positive assessment.

---

### Decision · Program_Chairs · 2026-04-30

**Decision:**

Accept (regular)

**Comment:**

This paper focuses on identifying the causal structure of partially observed linear causal models under heterogeneous environments. All reviewers have noted that the underlying problem is important and/or interesting.
Other strengths of this paper include novelty (Reviewers EH3k and fqW5), strong theoretical motivation through Theorem 1 (Reviewers b7XB, BMPm), and developments of hypothesis tests (Reviewers b7XB, BMPm and fqW5).

Reviewers b7XB, EH3k, and fqW5 were satisfied with the authors' rebuttals. Unfortunately, Reviewer BMPm did not engage in the later part of the review process. However, after reviewing the corresponding authors' rebuttal, I think the reviewer's questions and comments were addressed.

Given the strengths (significance of the underlying problem, novelty, and strong theoretical support) and the overall strong feedback from the reviewers, I recommend acceptance.